# Living fabrication of functional semi-interpenetrating polymeric materials

Zhuojun Dai[1,2✉], Xiaoyu Yang [3], Feilun Wu [2], Lihua Wang[2], Kun Xiang [2], Pengcheng Li[1], Qingqing Lv[1], Jinhui Tang[1,4], Anders Dohlman[2], Lei Dai[1], Xiling Shen [2,5] & Lingchong You [2,5,6✉]

Cell-mediated living fabrication has great promise for generating materials with versatile, programmable functions. Here, we demonstrate the engineering of living materials consisting of semi-interpenetrating polymer networks (sIPN). The fabrication process is driven by the engineered bacteria encapsulated in a polymeric microcapsule, which serves as the initial scaffold. The bacteria grow and undergo programmed lysis in a density-dependent manner, releasing protein monomers decorated with reactive tags. Those protein monomers polymerize with each other to form the second polymeric component that is interlaced with the initial crosslinked polymeric scaffold. The formation of sIPN serves the dual purposes of enhancing the mechanical property of the living materials and anchoring effector proteins for diverse applications. The material is resilient to perturbations because of the continual assembly of the protein mesh from the monomers released by the engineered bacteria. We demonstrate the adoption of the platform to protect gut microbiota in animals from antibiotic-mediated perturbations. Our work lays the foundation for programming functional living materials for diverse applications.

[1] Shenzhen Key Laboratory of Synthetic Genomics, Guangdong Provincial Key Laboratory of Synthetic Genomics, CAS Key Laboratory of Quantitative Engineering Biology, Shenzhen Institute of Synthetic Biology, Shenzhen Institutes of Advanced Technology, Chinese Academy of Sciences, Shenzhen, China. [2] Department of Biomedical Engineering, Duke University, Durham, NC, USA. [3] Systems, Synthetic, and Physical Biology Program, Rice University, Houston, TX, USA. [4] The Brain Cognition and Brain Disease Institute (BCBDI), Shenzhen Institutes of Advanced Technology, Chinese Academy of Sciences, Shenzhen, Guangdong, China. [5] Center for Genomic and Computational Biology, Duke University, Durham, NC, USA. [6] Department of Molecular Genetics and Microbiology, Duke University School of Medicine, Durham, NC, USA. ✉email: zj.dai@siat.ac.cn; you@duke.edu

An interpenetrating polymer network (IPN) is a combination of two or more polymers in networks that are interlaced with each other. If one of the polymers is not fully cross-linked, a semi-IPN (sIPN) results (Fig. 1a)[1–3]. By maintaining the key properties of each component, an engineered semi-IPN can be multi-functional[4,5]. For example, a semi-IPN composed of a temperature-sensitive component and a pH-sensitive component can respond to both cues[6]. Moreover, the interlacing between two or more components can reinforce or modulate the physical properties, such as mechanical strength of resultant material[7,8].

Although most semi-IPNs are assembled by the synthetic polymers, the incorporation of effector proteins can generate biological sensing or actuating capabilities. This incorporation can be achieved through physical absorption or entrapment of a functional protein to a scaffold, which, however, is prone to leaking due to the weak bonding[9]. Covalent bonding can enhance the immobilization of a target protein[10]. To do so, the latter has to be modified with multiple chemical synthesis steps to display chemically reactive groups. However, the harsh reaction environment may cause protein denaturation[9,11,12]. Also, the modifications may occur at multiple positions within a protein, making selective modification challenging and frequently yielding heterogeneous products[13].

Advances in protein engineering have made it possible to engineer protein tags that can form covalent bonds under mild conditions[14–16]. For example, protein polymers or hydrogels can be assembled by recombinant proteins with reactive tags[17,18]. However, this line of research is primarily done by using purified protein components, making the fabrication process time-consuming[17,19]. Importantly, materials assembled by chemical processes or purified proteins are sensitive to perturbations: after initial assembly, they cannot resume the lost function caused by perturbations.

The past 20 years have witnessed tremendous progress in synthetic biology, particularly in the ability to assemble complex gene circuits and to program certain well-defined dynamic functions. Recent examples have demonstrated the use of engineered bacteria to fabricate materials[20,21]. However, these studies have focused on a proof-of-concept demonstration of the fabrication process, instead of using the materials for specific applications. Also, the resulting materials do not exploit unique features of the living cells, such as sustained synthesis and biomolecules assembly, or the response to the environmental factors, although the engineered cells are critical for fabrication[20–22].

Here, we present a programmable approach to fabricate a biohybrid semi-IPN (sIPN) by co-opting engineered bacteria to grow the polymerized protein on top of a cross-linked scaffold. The core of the technology is a polymeric capsule encapsulating engineered bacteria that continually produce and release monomers to assemble the polymerized protein (Fig. 1b). The engineered bacteria can sense their physical confinement from the capsules and undergo autonomous lysis at a high local density. Upon lysis, the bacteria release monomers with multiple reactive tags, which can polymerize inside the capsules.

By design, the process is biocompatible and versatile to incorporate functional proteins by covalent bonds without requiring harsh reaction conditions. Due to the nature of the living fabrication, the material is resilient in its ability to recover its programmed function after transient or constant perturbations: i.e., it can self-repair during and post perturbations. Moreover, the encapsulated bacteria enable assembly of the protein polymer in a tunable manner. The formation of sIPNs leads to strengthening of the mechanical property of the proteins and the versatile functionalization of the scaffold polymer by incorporating different effector proteins, depending on the application context. To this end, we demonstrate the engineering of living materials to efficiently protect gut microbiota from an antibiotic-mediated perturbation.

## Results

**Fabrication of living sIPN capsules.** We used an engineered ePop circuit to program autonomous lysis of bacteria[23]. High cell densities lead to an increased plasmid copy number and greater E protein expression (a toxic protein interferes the cell wall synthesis), which then result lysis of a subpopulation of the bacteria and density-dependent-release of protein monomers. Unless noted otherwise, all bacteria described below are engineered to carry this circuit. Using this as the foundation, we further introduced circuits that encode different protein monomers. The backbone of the monomers is hydrophilic elastin-like polypeptides (ELPs)[17]. These ELPs were fused with either multiple SpyCatcher or SpyTag sequences, so they can polymerize by covalent bonding (Supplementary Fig. 1). SpyTag and SpyCatcher are a pair of reactive protein partners that can spontaneously react to reconstitute the intact folded CnaB2 domain under mild conditions[24]. The system is highly robust: the polymerization between a monomer containing SpyTags (T-mer) and one containing SpyCatchers (C-mer) can proceed efficiently at biocompatible environmental conditions[15,25]. We used the chitosan to assemble the initial polymeric scaffold[26]. The positive-charged amine groups of chitosan are cross-linked by non-toxic polyanion negative-charged tripolyphosphate (Supplementary Fig. 2). The scaffold traps the living cells and provides a concentrated monomers environment, facilitating covalent bonds formation between the monomers (Fig. 1b).

We generated four genes encoding four basic monomers: two T-mers and two C-mers (Supplementary Fig. 1). A plasmid containing each gene under control of an inducible promoter (Supplementary Materials) and the ePop circuit were co-transformed into MC4100Z1 *E. coli* bacteria (MC). Thus, each resulting strain can produce and release a single monomer through programmed autolysis. We first co-cultured bacteria producing T-mers (MC ($T_2$-mCherry) or MC ($T_3$)) and bacteria producing C-mers (MC($C_2$) or MC ($C_3$)) in the liquid culture, harvested and purified the supernatant after 24 h. SDS–PAGE gel indicated the formation of protein complexes

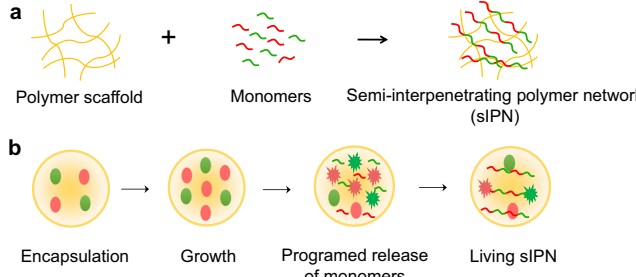

**Fig. 1 Conventional and living fabrication of semi-IPN. a** A semi-interpenetrating polymer network (sIPN) is defined as a linear or branched polymer in the presence of another cross-linked polymer. sIPN can be assembled by dissolving monomers (for the 2nd component) in a cross-linked scaffold (the 1st component), before initiating polymerization. **b** Living fabrication of sIPN using engineered bacteria. When encapsulated inside a polymeric microcapsule (the 1st component, cross-linked), the engineered bacteria can undergo autonomous lysis at a high local density. Lysis releases protein monomers that can react to form the polymerized protein (the 2nd component, polymerized but not necessarily fully cross-linked) inside the polymeric scaffold.

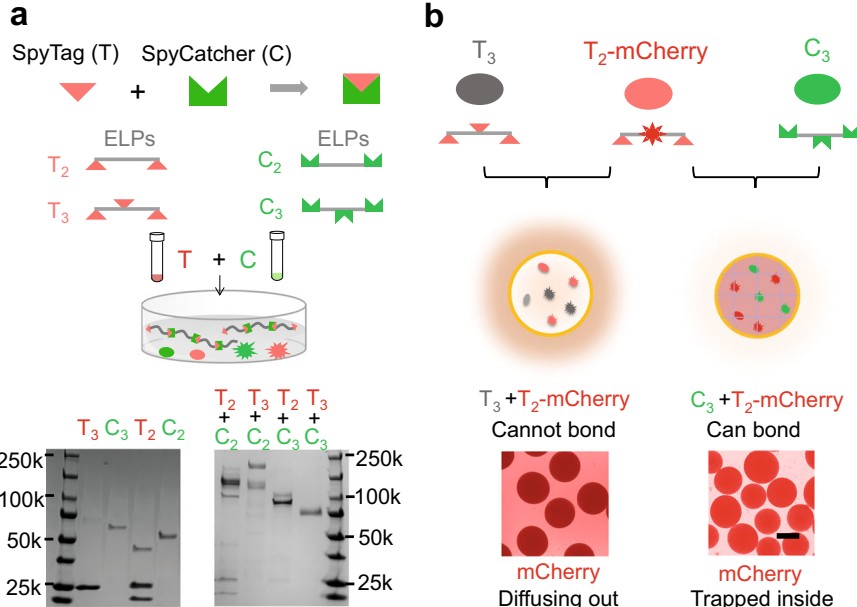

**Fig. 2 Formation of living sIPN. a** Formation of protein complexes by elastin-like polypeptides (ELPs) monomers containing SpyTags and SpyCatchers released from engineered bacteria (top: schematic; bottom: experimental data). ELPs decorated with multiple SpyTags or SpyCatchers released by the co-cultured bacteria reacted to form covalent bonds in the supernatant. Overnight culture of bacteria expressing proteins with multiple SpyTags (T, MC($T_2$-mCherry) or MC($T_3$)) and bacteria expressing proteins with multiple SpyCatchers (C, MC($C_2$) or MC($C_3$)) were co-cultured in M9 medium containing 1 mM IPTG. After 24 h, the supernatant was harvested, purified by His-tag affinity resins and tested by SDS–PAGE. Compared with the monomers (left), all four combinations between SpyTag and SpyCatcher generated protein complexes with higher molecular weight, indicating reaction occurred between monomers. The experiment was repeated more than three times independently with similar results. **b** Stronger mCherry signal inside living sIPN capsules containing compatible monomers released by engineered bacteria (top: schematic; bottom: experimental data). MC($T_2$-mCherry) and MC ($C_3$) were mixed, pelleted and encapsulated with chitosan. As a control, MC($C_3$) was replaced with MC($T_3$). The capsules were cultured in M9 medium containing 1 mM IPTG. Due to the sIPN formation, the mCherry protein was immobilized inside the capsules, leading to strong fluorescence of the capsules but little in the surrounding medium (right). In the control group, the mCherry protein was not trapped due to the lack of sIPN formation, and diffused out to the surrounding medium (left). The scale bar is 200 µm and the photos were taken at 24 h. The experiment was repeated more than three times independently with similar results.

by the monomers, as the presence of bands at higher molecular weight (Fig. 2a).

In a liquid culture, the monomers are diluted upon being released from lysed bacteria. In contrast, microcapsules can concentrate the monomers and facilitate polymerization. Using the same number of cells, the encapsulation (~400 µL) can provide ~10 times concentrated monomer environment than liquid culture (4 mL nutrients). The concentration of monomers in each capsule increased to ~7 µM after 12 h, corresponding to an increase in the cell number (Supplementary Fig. 3). At such a concentration, the monomers can react with a half-life of several minutes[25], and polymerize more efficiently (>460 kDa for $T_2$-mCherry and $C_3$) compared with the reaction in the supernatant, as indicated by the PAGE gel (Supplementary Fig. 4).

To fabricate living sIPN capsules, we mixed bacteria producing a T-mer containing two SpyTags fused to mCherry as a reporter (MC ($T_2$-mCherry)) and bacteria producing a C-mer containing three SpyCatchers (MC ($C_3$)) and encapsulated them with chitosan. As a control, MC ($C_3$) was replaced with MC ($T_3$), which produces a T-mer containing three SpyTags. Since $T_3$ and $T_2$-mCherry cannot form covalent bonds, they do not polymerize inside capsules (non-IPN capsules). As such, we expect $T_2$-mCherry to leak out. Consistent with this notion, the culture containing non-IPN capsules exhibited substantial red fluorescence outside the capsules (Fig. 2b, left; Supplementary Movie 1). In contrast, living sIPN capsules had much brighter red fluorescence inside the capsules, with little leakage (Fig. 2b, right; Supplementary Movie 2). During growth, the bacterial culture changed the growth environment in multiple aspects, including

the pH and ionic strength, and therefore caused the shrinking of chitosan capsules[27]. This phenomenon is more pronounced in non-IPN capsules, possibly because the polymerized protein components (hydrophilic ELPs) stay swollen and restrain the polymeric network from collapsing in living sIPN capsules. We further washed both capsules, dehydrated, sectioned, and imaged by confocal microscopy. The living sIPN capsules show overall stronger mCherry signal, which is consistent with what we observed from movies (Supplementary Fig. 5). SEM images indicate smaller pore size inside living sIPN capsules, compared with non-IPN capsules (Supplementary Fig. 6). To observe the polymerized protein inside living sIPN, we dehydrated, embedded, and sliced both non-IPN and living sIPN capsules into thin sections. The sections were placed on a coated copper grid and stained with Ni-NTA-AuNPs (gold nanoparticles, 5 nm). Since the protein monomers were His-tagged, the polymerized protein can be differentiated due to its conjugation with AuNPs. Our results confirm the existence of polymerized protein inside living sIPN capsules as indicated by the densely alignment of the AuNPs. In comparison, there is only sparse and scattered distribution of AuNPs inside non-IPN capsules (Supplementary Fig. 7).

The formation of sIPN can strengthen the resulting material. To test this notion, we performed dynamic shear rheology experiments in the strain sweep and frequency-sweep mode, in which storage modulus was monitored for living sIPN, non-IPN capsules, capsules encapsulating cells but not producing any monomers and capsules containing no cells. The storage modulus of the capsules containing no cells stayed constant after 24 h. Both

non-IPN capsules and capsules encapsulating cells but not producing any monomers increased by ~2-fold during the same time window, likely due to the cell growth. In contrast, the storage modulus of living sIPN capsules increased by ~4-fold, suggesting an additional role in the formation of a semi-interpenetrating polymeric network (Supplementary Fig. 8).

**Bla as a model protein for functional living sIPN fabrication.** Depending on the application context, different functional proteins can be incorporated into the living sIPN material in a flexible manner due to its programmability. Here, we chose the beta-lactamase (Bla) as a model protein to functionalize the living sIPN capsules. We used these functional living sIPN capsules to minimize the perturbation of the microbiota during blood-infusion of a beta-lactam antibiotic.

Intravenous (IV) administration of antibiotics into the bloodstream is often used to treat severe bacterial infections[28]. Beta-lactams account for 72% of IV-administrated antibiotics in the United States annually[29]. After injection, the antibiotics rapidly accumulate in the bile and in the gut subsequently[30]. This unintended accumulation can cause severe disruption of the normal intestinal microflora, increasing the susceptibility of a patient to secondary infections caused by pathogens such as *Clostridium difficile*[31–34]. One approach to protect the microbiome from antibiotic-mediated perturbation is to use Bla to degrade beta-lactams in the gastrointestinal tract[29,35,36]. However, free Bla is prone to breakdown (e.g., by proteinases) inside the gut, reducing the efficiency of protection[30,36,37]. We reason that our platform can increase the treatment efficacy because of the continual production of Bla by the living bacteria as well as the enhanced Bla stability due to the formation of sIPN (Supplementary Fig. 9a)[38,39].

To fabricate living sIPN capsules containing Bla (living Bla sIPN capsules), we mixed bacteria producing a T-mer containing three SpyTags fused to Bla (MC (T$_3$-Bla)) and bacteria producing a C-mer containing three SpyCatchers (MC(C$_3$)), and encapsulated them with chitosan. As a control, we manufactured living non-IPN capsules producing Bla by replacing MC(C$_3$) with MC (T$_3$). Both capsules were cultured in M9 medium and harvested at various time points for Bla activity evaluation. As shown in Fig. 3a, the Bla activity of both capsules accumulated gradually over time, likely due to the continual production by the engineered bacteria. At 24 h, the Bla activity of living Bla sIPN capsules (anchoring ~10$^{10}$ Bla molecules per capsule, see "Methods") was ~3.27-fold of that of non-IPN capsules, indicating a greater retention of Bla in sIPN-mediated immobilization.

The continual synthesis by the engineered bacteria can also confer the living sIPN material resilience to perturbations. That is, the material has the ability to recover its function after transient or prolonged perturbations. Such a property can facilitate robust function of the material in a complex environment, where multiple factors may transiently deactivate the material. To test this notion, we supplemented clavulanic acid, an effective Bla inhibitor[40], to supernatant containing Bla of the liquid culture or the living Bla sIPN capsules for 10 min or 24 h as a transient or prolonged perturbations. When clavulanic acid was supplemented for 24 h (co-cultured with 0.1, 0.5, or 1 μg/mL clavulanic acid overnight), living Bla sIPN capsules maintained ~65–86% of the enzymatic activity; in contrast, living non-IPN capsules producing Bla maintained ~9–45% activity, while the Bla activity in the supernatant was completely quenched (Fig. 3b and Supplementary Fig. 10b). When clavulanic acid was supplemented for 10 min, the living Bla sIPN capsules partially stabilized the Bla function and maintained ~66, 45, or 11% (~38%, 29%, or zero

for living non-IPN capsules producing Bla) of the enzymatic activity when treated by 0.1, 0.5, or 1 μg/mL clavulanic acid (Supplementary Figs. 9b and 10a; Supplementary Table 1). All the clavulanic acid treated living Bla sIPN capsules fully resumed the activity after another 24 h culture. In comparison, following the same treatment, living non-IPN capsules producing Bla restored ~53–87% of enzymatic activity, and the supernatant (containing Bla before treatment) or capsules containing pure Bla did not regain any enzymatic activity (Supplementary Figs. 9b, 10a, and 10c).

To test the protection of microbiome by the functionalized living material, we transformed our circuit into a probiotic *E. coli* strain, Nissle 1917 (NI), which is widely considered to be safe for human consumption[41,42]. We first confirmed the ePop function and expression of protein monomers in NI (Supplementary Fig. 11). We then encapsulated NI(T$_3$-Bla) with NI(C$_3$) using chitosan to manufacture the living Bla sIPN capsules. As control, we manufactured living non-IPN capsules producing Bla by replacing NI(C$_3$) with NI(T$_3$). Consistent with the results using MC4100Z1 strain (Fig. 3), sIPN fabricated by Nissle strain can anchor the Bla and enhance the enzyme stability in response to perturbations (Supplementary Fig. 12a and b). Chitosan scaffold can further protect the living cells from acidic condition, mimicking the pH in the stomach, since the amine groups are able to buffer low pH by protonation. In the liquid culture, the number of viable bacteria decreased from ~10$^9$ to 10$^3$ CFU/mL after being exposed to a highly acidic environment (pH = 1) for 1 h. In contrast, the number of viable encapsulated bacteria decreased from ~10$^8$ to 10$^6$ CFU/mL after the same treatment (Supplementary Fig. 13). We further tested if the treated cells could re-grow and function properly by culturing the cells (protected or unprotected) in the medium for another 24 h. Our results showed that the Bla activity produced and released by the protected cells (treated in the acidic environment for 2 h) was ~90% of the activity under the untreated condition. These results indicate the maintenance of the living-synthesis capability by the encapsulated bacteria (Fig. 3c).

We randomly divided 40 six-week-old BALB/c male mice into four groups ($n = 10$ for each group). The mice were kept in the animal facility for 1 week to stabilize the gut microbiome before antibiotics treatment. On day 0, feces samples were collected from all mice and used as a reference point. From days 1–3 (on each day), three different capsules (25,000 capsules for each mouse) were fed (by oral gavage) to mice of groups 2–4 one hour before 5 mg of ampicillin was injected into each mouse through the tail vein (Fig. 4a)[35]. Mice of group 1 (no capsules) received only antibiotic injection. Living Bla sIPN capsules and living non-IPN capsules containing bacteria producing Bla were administered to mice of groups 2 (living sIPN) and 3 (living non-IPN), respectively. Both types of capsules were cultured in M9 medium for 20 h before being fed to mice. Pure Bla capsules (capsules containing purified T$_3$-Bla but no cells) were administered to mice of group 4 (pure Bla). The initial Bla activity of pure Bla capsules was ~2.5 times higher than that of living Bla sIPN capsules. In addition to day 0, feces samples were also collected on days 4, 7, and 14.

We used several different measurements to quantify perturbations of gut microbiome over time. We used fecal DNA content to estimate the absolute abundance of gut microbiota between day 0 and each of the following time points (days 4, 7, and 14) for all 40 mice individually (Fig. 4a). We further used 16S rRNA sequencing to track changes in the composition of gut microbiota throughout the antibiotics-perturbation experiment (Fig. 4a, b and Supplementary Fig. 14). The weighted UniFrac takes into account both phylogenetic lineages and relative abundance of two samples[43]. The higher the weighted UniFrac distance, the more

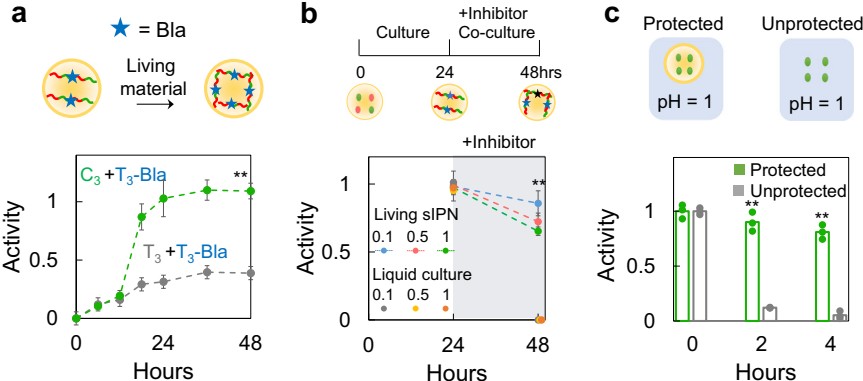

**Fig. 3 Living sIPN enhanced or stabilized anchored functions. a** sIPN formation enhanced the activity of the anchored enzyme (Bla) over time (top: schematic; bottom: experimental data). MC($T_3$-Bla) and MC($C_3$) bacteria were mixed, encapsulated with chitosan, and cultured in M9 for 48 h. As a control, MC ($C_3$) was replaced with MC ($T_3$). The capsules were harvested at different time points and washed by PBS before enzymatic assay. The y-axis indicates the Bla activity in the unit of $OD_{490}$/h. The experiments were done in triplicate. At 48 h, the activity from sIPN capsules was ~2.8 times that of non-IPN capsules (**$p < 0.01$ ($p = 0.0002$), determined by the two-sided Student's two-sample $t$-test assuming unequal variances; error bars = standard deviation ($n = 3$)). **b** Living sIPN capsules maintained enzymatic activity under sustained perturbation (top: schematic; bottom: experimental data). Cells (MC($T_3$-Bla)) and cells (MC($C_3$)) were mixed and inoculated into M9 (liquid culture), or encapsulated with chitosan (living sIPN capsules) and cultured in the same amount of M9 for 24 h. The samples (the supernatant containing Bla for the liquid culture, and the capsules for living sIPN capsules) were collected and sent for Bla enzymatic assay. Then, the clavulanic acid (0.1, 0.5, or 1 µg/mL) was added to the liquid culture, or living sIPN capsules for another 24 h (shaded area) before the Bla enzymatic assay. The data were normalized by the control (liquid culture or living sIPN capsules not treated by clavulanic acid). The experiments were done in triplicate. At 48 h, the enzymatic activity of living sIPN capsules was significantly greater than that of the liquid culture, which was close to be zero (**$p < 0.01$ ($p = 0.004$, 0.003 and 0.001 for 0.1, 0.5, or 1 µg/mL clavulanic acid concentration), determined by the two-sided Student's two-sample $t$-test assuming unequal variances; error bars = standard deviation ($n = 3$)). **c** Chitosan capsules protected bacteria from the acidic environment (top: schematic; bottom: experimental data). Unprotected or encapsulated NI ($T_3$-Bla) bacteria were incubated in the pH = 1 buffer for 0 (control), 2 or 4 h, respectively. After incubation, the unprotected bacteria and capsules were collected and cultured in M9 medium for another 24 h. The supernatant was collected and subject to Bla enzymatic activity assay. The data indicate the Bla activity normalized by control (treated by pH = 1 buffer for 0 h). The bar represents the mean value ($n = 3$). The experiments were done in triplicate. The activity under the protected condition was significantly greater than that under the unprotected condition (**$p < 0.01$ ($p = 0.004$ and 0.0004 for 2 and 4 h), determined by the two-sided Student's two-sample $t$-test assuming unequal variances).

dissimilar is a gut microbiome composition from that of day 0 (Fig. 4a). Our results revealed that the gut microbiota of mice in group 1 (no capsules) on day 4 were severely perturbed after being treated with ampicillin (days 1–3), in terms of the absolute abundance and the taxonomy profile of gut microbiota. In comparison, treatment by using various capsules that contain Bla all reduced the perturbation of ampicillin on gut microbiota to varying degrees (Fig. 4). Though living Bla sIPN capsules (group 2) had less initial Bla activity than capsules containing pure Bla (group 4), living sIPN capsules exhibited the most potent protection, especially during the early stage (i.e., day 4) (Supplementary Table 2). This result underscores the benefit of living synthesis and sIPN-mediated stabilization of Bla (Fig. 4 and Supplementary Fig. 14). Living non-IPN capsules containing bacteria producing Bla (group 3) demonstrated comparable protection as capsules containing pure Bla only. All three types of capsules stayed inside the gut for 4–5 h before cleared out of the body (Supplementary Fig. 15).

## Discussion

We have demonstrated a new approach to fabricate semi-IPN that is living, functional and biocompatible. In comparison with conventional methods, living fabrication does not require the labor-intensive processes of preparing the purified components or extensive optimization of the fabrication process. The polymerized protein component in the semi-IPN is assembled by the monomers produced and released by the engineered autolysis bacteria. Depending on specific application contexts, the effector proteins can be released by secretion, as demonstrated in a recent

study[44]. Due to the circuit design, the living functional material is resilient to perturbations and robust in maintaining the programmed functions. The system is modular: the engineered gene circuit, the bacterial strain, the functional protein, the polymerization chemistry, and encapsulating material can be separately optimized and then integrated. For example, the SpyTag-SpyCatcher chemistry can be optimized or substituted with other reaction systems for protein polymerization. We used chitosan as encapsulating material partially due to its amine groups can be protonated to protect the cells at acidic environment. The system can be further modified to adjust their circulation dynamics and cell escape rate by surface modifications (Supplementary Figs. 16 and 17). Besides micro-sized capsules, our strategy could be extended to build bulk living material due to its modularity. For example, we can mix alginate with engineered bacteria and cross-link the system with calcium ions to form bulk gel[45,46]. The bacteria inside the bulk gel assemble the polymerized protein, which then interlaces with the alginate networks to form the semi-IPN.

Our system demonstrates the integration of synthetic biology, protein engineering, and material science to achieve living functional material synthesis. In particular, it represents the first example to use living cells to create active materials, of which the function also requires programmed behavior of the cells. Our work is also the first example where bacterially fabricated living materials are used in a setting with direct clinical relevance—the protection of the gut microbiota against unintended perturbations by antibiotics, and the unique features of the living materials are critical for the target application. Variants of our system can be adopted for other types of applications, including controlled drug release, environmental cleanup, or self-healing material.

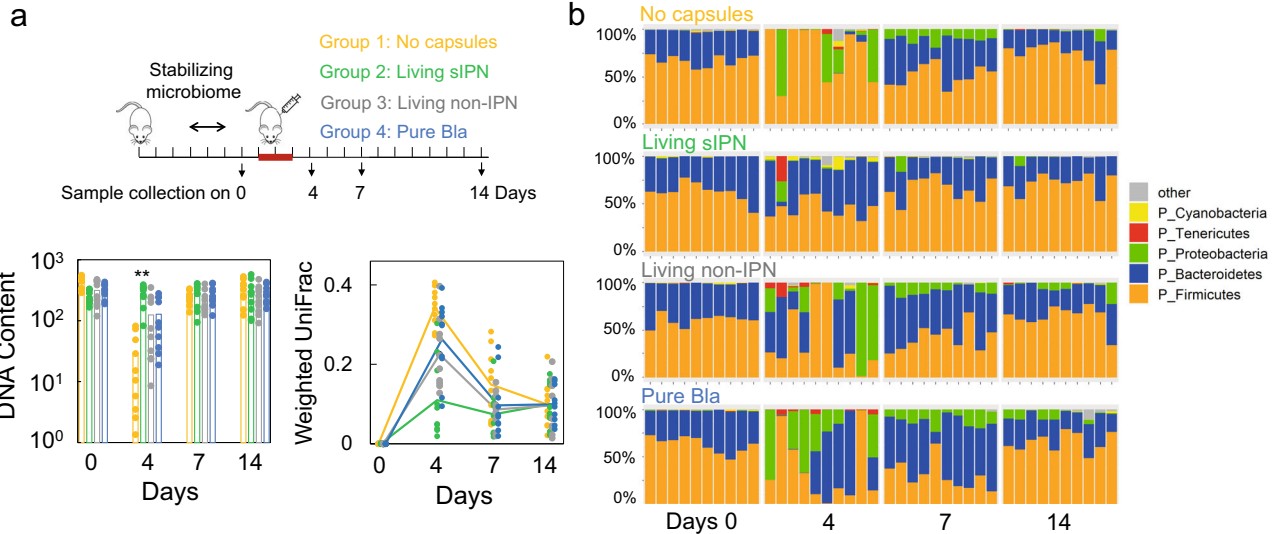

**Fig. 4 Living Bla sIPN capsules minimized antibiotic-mediated perturbations of the gut microbiome in mice. a** Living Bla sIPN capsules provided the most potent protection of the gut microbiome from antibiotic-mediated perturbations (top: experimental setup; bottom: experimental data). Forty mice were separated into four groups and kept in the facility for 1 week to stabilize the gut microbiome. Fecal samples were collected on day 0 and used as reference point to evaluate the disturbance to the microbiome. The treatment was given from day 1 to 3. On each day, living Bla sIPN capsules, living non-IPN capsules producing Bla or pure Bla capsules were administered to groups 2–4, respectively. After 1 h, 5 mg ampicillin (100 μL) was injected into each mouse through the tail vein for all four groups. Fecal samples from 40 mice were collected on days 0, 4, 7, 14 for DNA quantification and 16S rDNA sequencing. The left panel shows the DNA content (ng DNA per mg fecal matter). Each dot represents an individual mouse, and the bar represents the average value ($n = 10$). The right panel shows the weighted UniFrac distance between the gut microbiome at days 4, 7, 14, and the reference point (day 0) as analyzed by QIIME2 (see "Methods"). The weighted UniFrac distance takes into account both phylogenetic lineages and relative abundance of two samples and is commonly used as the distance metric for microbial communities. The line represents the mean value ($n = 10$). On day 4, living sIPN capsules exhibited the most potent protection in comparison to the other three groups, in terms of maintaining the total DNA content and microbiome composition (**$p < 0.01$ ($p = 10^{-5}$, 0.005 and 0.003 for group 2 compared with groups 1, 3, and 4 on day 4), determined by the two-sided Student's two-sample $t$-test assuming unequal variances). The statistical analysis of the weighted UniFrac was shown in Supplementary Table 2. **b** The relative abundances of major phyla for mice gut microbiota pretreated with different capsules or not pretreated before antibiotic injection. Each row represents a different treatment group. Each column in the panel corresponds to an individual mouse. The $x$-axis represents the numbering of the mice, and $y$-axis represents the relative abundance of different phyla. Typically, in the absence of perturbations, Bacteroidetes (blue) and Firmicutes (orange) dominate the mice gut microbiome. In living Bla sIPN group, these two phyla remained dominant and the microbiome composition remained stable (Supplementary Fig. 14). In the absence of pre-treatments, the relative abundance of these major phyla was drastically altered following antibiotics injection.

## Methods

**Overnight liquid culture**. Bacteria carrying the ePop circuit were streaked onto an agar plate supplemented with 2% (w/v) glucose and incubated at 37 °C for 16 h. Then, a single colony was picked and inoculated into LB media with 2% (w/v) glucose and appropriate antibiotics when applicable in the overnight cultures. Bacteria carrying inducible protein-expression circuits (T5) were induced with 1 mM IPTG.

**Production of chitosan microcapsules containing bacteria**. Two percent (w/v in 1% acetic acid solution) chitosan (Sigma-Aldrich) solution and bacteria solution were mixed and loaded into a syringe attaching with a blunt tip. The syringe was placed onto a syringe pump, with the needle connected to the anode of a high voltage DC power supply (Dongwen high voltage) and a sterile metal receiving container (loaded with cross-linking solution 5% (w/v) tripolyphosphate, Sigma-Aldrich) connected to the cathode. The mixture of chitosan and bacteria solution was sprayed at 5 kV to the receiving container with stirring. The microcapsules were collected by centrifugation at $500 \times g$ for 5 min and then washed twice by PBS.

**Protein purification**. For His-tag affinity purification, we used 75 μL cOmplete™ His-Tag purification resin (Sigma-Aldrich) to bind 4 mL supernatant and eluted the resin by 75 μL elution buffer. For all protein samples, SDS–PAGE with Coomassie Blue stain was used to verify the protein monomers and polymerized proteins.

**Preparation and observation of capsules and frozen sections**. To fabricate living sIPN capsules, 187.5 μL overnight culture of MC (T$_2$-mCherry) and 187.5 μL overnight culture of MC (C$_3$) were mixed, centrifuged and re-suspended in 37.5 μL LB medium, and encapsulated into 375 μL chitosan solution. The capsules were inoculated into 4 mL M9 with appropriate antibiotics and 1 mM IPTG and cultured at 37 °C. To fabricate non-IPN capsules, MC (C$_3$) was replaced with MC (T$_3$), while other procedures were the same. An inverted fluorescence microscope (Nikon Eclipse Ti) with ×10 objective was employed to monitor the sIPN formation. ImageJ was used to generate the supplementary movies.

To obtain the frozen sections, sIPN and non-IPN capsules (cultured for 24 h) were first dehydrated in the 30% sucrose solution for 6 h. The capsules were then embedded using tissue freeze medium (SAKURA Tissue-Tek® O.C.T.) and frozen in the −80 refrigerator overnight. The resultant system was then cut to 10-μm thick tissue sections using a cryostat (Leica CM3050 S). The section was thaw-mounted on a coverslip for confocal microscopy observation (Nikon A1R).

**Scanning electron microscope (SEM) observation**. The procedures to prepare chitosan capsules, living sIPN, and non-IPN capsules were described as above. All capsules were cultured for 24 h in media. Chitosan capsules, living sIPN and non-IPN capsules were first frozen in liquid nitrogen, and then lyophilized overnight (Christ Alpha1-2LDplus). The capsules were dissected and then observed under the SEM (Phenom XL).

**Transmission electron microscopy (TEM) observation**. The procedures to prepare living sIPN and non-IPN capsules were described as above. All capsules were cultured for 24 h in media. Living sIPN and non-IPN capsules were first dehydrated in the 30% sucrose solution for 6 h. The capsules were then embedded, thin sectioned using a cryostat (Leica CM3050 S), and mounted onto coated grids. The grids were then washed by 30 μL selective binding buffer (20 mM Na$_2$HPO$_4$, 500 mM NaCl, 20 mM imidazole, pH = 7.4) twice, and placed upside down on a droplet containing Ni-NTA-AuNP (10 nM, Nanoprobes) and incubated for 90 min at room temperature. The grids were then washed twice with selective binding buffer and twice with D.I. water. Before imaging, the grids were negatively stained by 10 μL of 2% (w/v) phosphotungstic acid solution, washed twice by D.I. water, followed by air-dried and examined by a FEI Spirit T12 transmission electron microscope. Images were taken with an Orius SC200B 200 kV camera.

**Dynamic shear rheology measurement**. Dynamic strain-, and frequency-sweep experiments were performed on a TA Instruments ARES-RFS strain-controlled rheometer with a standard steel parallel-plate geometry (8-mm diameter). A layer

of capsules was dispersed on the plates and was monitored by strain sweep and frequency sweep, respectively, at 25 °C. Strain sweep was performed from 0.1 to 10% at a fixed frequency of 10 rad/s at 25 °C. Frequency sweep was performed from 10 to 0.1 Hz by holding the strain at 5%.

**Preparation of living Bla sIPN capsules**. To fabricate living Bla sIPN capsules, 312.5 μL overnight culture of MC (T$_3$-Bla) and 62.5 μL overnight culture of MC (C$_3$) were mixed, centrifuged and re-suspended in 37.5 μL LB medium, and encapsulated into 375 μL chitosan solution. The capsules were inoculated into 4 mL M9 with appropriate antibiotics and 1 mM IPTG and cultured at 37 °C for 24 h. To fabricate living non-IPN capsules producing Bla, MC (C$_3$) was replaced with MC (T$_3$), while other procedures were the same.

**Measuring Bla activity**. The Bla activity was quantified by nitrocefin (Abcam) assay. 5 μL supernatant or 5 μL capsules were diluted into 100 μL using PBS and mixed with substrate nitrocefin (concentration = 50 μM). The resulting absorbance at 490 nm due to the generation of a colored produce was measured as a function of time by a platereader (BioTek Epoch 2). The measurement of enzymatic activity can be explained by the Michaelis–Menten kinetics described in our previous paper[27]. Experimentally, enzymatic activity is calculated by taking the time derivative of the absorbance at 490 nm in the initial time window, when the absorbance increases linearly with time.

To quantify Bla activity in one capsule, we used the purified T$_3$-Bla to first generate a calibration curve (Supplementary Fig. 3a). We then measured the enzymatic activity for capsules and back-calculated the concentration of Bla in capsules ($C_{capsules}$/μM). Each capsule anchoring Bla was calculated based on $M_{each\ capsule} = C_{capsules} \times V_{capsuels} \times N_A$.

**Evaluation of cell number and monomer concentration inside capsules**. In total, 375 μL of overnight culture of MC(T$_3$-Bla) were centrifuged and re-suspended in 37.5 μL LB medium and encapsulated into 375 μL chitosan solution. The capsules were inoculated into 4 mL M9 with appropriate antibiotics and 1 mM IPTG. To estimate the number of monomers inside the capsules, we cultured these capsules and collected the capsules (without washing) at different time points. We released the monomer from the capsules by treating with 2.5 M NaCl (vortexing 5 min) followed by PBS (vortexing 10 min). The Bla activity in the supernatant was measured and then back-calculated to the concentration in capsules (Supplementary Fig. 3b).

To measure the cell viability in the encapsulation, we made the capsules encapsulating MC(T$_3$-Bla) (same procedures above), cultured in 4 mL M9 and collected the capsules at different time points. To release bacteria from capsules, we used 2.5 M NaCl (vortexing 5 min) followed by PBS (vortexing 10 min) to destabilize the capsules. Colony-forming unit (CFU) was used to estimate the number of viable bacteria (Supplementary Fig. 3c).

**Evaluation of transient or prolonged perturbation on sIPN capsules**. To fabricate living Bla sIPN capsules, 312.5 μL overnight culture of MC (T$_3$-Bla) and 62.5 μL overnight culture of MC (C$_3$) were mixed, centrifuged and re-suspended in 37.5 μL LB medium, and encapsulated into 375 μL chitosan solution. The capsules were inoculated into 4 mL M9 with appropriate antibiotics and 1 mM IPTG and cultured at 37 °C for 24 h. To fabricate living non-IPN capsules producing Bla, MC (C$_3$) was replaced with MC (T$_3$), while other procedures were the same. For liquid culture, 312.5 μL overnight culture of MC (T$_3$-Bla) and 62.5 μL overnight culture of MC (C$_3$) were mixed and inoculated into 4 mL M9 with appropriate antibiotics and 1 mM IPTG and cultured in 37 °C for 24 h.

For transient perturbations, clavulanic acid (0.1, 0.5, or 1 μg/mL, Dr. Ehrenstorfer GmbH) was added to the supernatant of the liquid culture, living sIPN capsules or non-IPN capsules and incubated for 10 min before the Bla assay. For liquid culture, the Bla supernatant was first purified using His-tagged beads to remove the inhibitor, re-inoculated into the same amount of M9 and cultured for another 24 h before the enzymatic assay. For living sIPN or non-IPN capsules, the treated capsules were collected and re-inoculated into the same amount of M9 nutrients and cultured for another 24 h before the enzymatic assay.

For prolonged perturbation, after 24 h culture, the Bla supernatant was first purified using His-tagged beads, re-inoculated into the same amount of M9 with clavulanic acid (0.1, 0.5, or 1 μg/mL) and cultured for another 24 h before the enzymatic assay. For living sIPN or non-IPN capsules, the capsules were collected and re-inoculated into the same amount of M9 nutrients with clavulanic acid (0.1, 0.5, or 1 μg/mL) and cultured for another 24 h before the enzymatic assay.

**Monitoring circulation dynamics of capsules inside mice**. 375 μL of overnight culture of NI (Luminescence) without ePop were centrifuged and re-suspended in 37.5 μL LB medium and encapsulated into 375 μL chitosan solution. These capsules were inoculated into 4 mL M9 with appropriate antibiotics and 1 mM IPTG and cultured at 37 °C for 20 h. The capsules were washed by PBS before administrated to the mice by gavage. Around 25,000 capsules were administrated to each mouse. The mice were anaesthetized and imaged at different time points by an IVIS Lumina III In Vivo Imaging System. We have complied with all relevant ethical regulations during the animal experiments.

**Modification on the surface of the capsules by layer-by-layer (LbL) coating technique**. 375 μL capsules were washed first with PBS, and then incubated in 1.5 mL alginate solution (0.1%/(w/v)) for 15 min with gentle shaking. The resultant capsules were centrifuged, collected, washed again with PBS, and then incubated in 1.5 mL chitosan solution (0.4%/(w/v)) with gentle shaking for another 15 min. The resultant capsules were washed again with PBS and collected.

**Treatment on mice with various capsules and feces samples collection**. To fabricate living Bla sIPN capsules, 1.67 mL overnight culture of NI (T$_3$-Bla) and 0.33 mL overnight culture of NI (C$_3$) were mixed, centrifuged, and re-suspended in 0.2 mL LB medium, and encapsulated into 2 mL chitosan solution. The capsules were cultured in M9 with appropriate antibiotics and 1 mM IPTG and cultured at 37 °C for 20 h. To fabricate living non-IPN capsules producing Bla, MC (C$_3$) was replaced with MC (T$_3$), while other procedures were the same. T$_3$-Bla was purified and mixed with chitosan to generate capsules containing pure Bla.

Male BALB/c mice, aged 6 weeks, were obtained from The Vital River Laboratories (Beijing, CN). Mice were housed under standard conditions with a 12 h light–dark cycle at 20–26 °C (daily temperature difference <4 °C), humidity 40–70%, and free access to food (GB 14924.3-2010 feed formula) and water. We randomly divided 40 six-week-old BALB/c male mice into 4 groups ($n = 10$ for each group). The mice were kept in the animal facility for 1 week to stabilize the gut microbiome before antibiotics treatment. On day 0, feces samples were collected from all mice and used as a reference point. From days 1–3 (on each day), three different capsules (200 μL, ~25,000 capsules for each mouse) were fed (by oral gavage) to mice of groups 2–4 one hour before 5 mg of ampicillin was injected into each mouse through the tail vein.

To collect feces sample, individual mouse was placed in a clean empty cage according to the mouse number. The mouse was returned to the original feeding cage when sampling requirements are met. The fecal pellets were collected in cryopreservation tubes, snap-frozen in liquid nitrogen, and transferred to storage at −80 °C. All animal experiments were approved by the Institutional Animal Care and Use Committee at Shenzhen Institutes of Advanced Technology (permit number: SIAT-IACUC-191230-HCS-DL-A0979).

**16S rRNA sequencing and data analysis**. DNA extraction from the mice fecal was performed according to the instructions of the PowerFecal Pro DNA Kit (QIAGEN) and eluted in 100 μL elution buffer provided in the kit. The DNA content was quantified using NanoDrop 2000c (Thermo Scientific). Sequencing libraries were generated using NEBNext Ultra™ II DNA Library Prep Kit for Illumina (New England Biolabs, MA, USA) following manufacturer's recommendations and index codes were added. The library quality was assessed on the Qubit 2.0 Fluorometer (Thermo Fisher Scientific, MA, USA). At last, the library was sequenced on an Illumina Nova6000 platform and 250 bp paired-end reads were generated (Guangdong Magigene Biotechnology Co., Ltd., Guangzhou, China).

Alignment and demultiplexing of raw 16S ribosomal RNA sequencing data was performed with QIIME2 (version 2019.7)[47]. Primers of the raw sequence data were cut with Cutadapt (via q2-cutadapt)[48], followed by denoising, merged and removing chimera with DADA2 (via q2-dada2)[49]. All amplicon sequence variants (ASVs) from DADA2 were used to construct a phylogeny with fasttree2 (via q2-phylogeny)[50]. The taxonomy of ASVs was assigned by naïve_bayes classifier (via q2-feature-classifier)[51] against the silva database (SILVA_138_SSURef_Nr99). The ASV table was normalized by the sample with the fewest sequence reads, and rare ASVs (<0.1% in relative abundance) were filtered out. Weighted UniFrac distance was calculated by the Vegan package in R (https://CRAN.R-project.org/package=vegan).

**Reporting summary**. Further information on research design is available in the Nature Research Reporting Summary linked to this article.

## Data availability
The authors declare that the source data processed for figures generation in this study are available within the paper and its Supplementary files. Any additional information is available upon request. Source data are provided with this paper.

## Code availability
The code that support the findings of this study are available from the corresponding author upon reasonable request.

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

## Acknowledgements

We thank Tatyana A. Sysoeva and Nan Luo for insightful comments and suggestions. We thank Yixuan Chen, Ying Wang, Runtao Zhu, Xi Zhang, Weiqi Lu, Jing Sun, and Yang Zhao for help in experimental setup during the manuscript revision. We thank Testing Technology Center of Materials and Devices, Tsinghua Shenzhen International Graduate School for TEM instrument usage. This study was partially supported by National Key Research and Development Program of China No. 2018YFA0903000 and No. 2020YFA0908100 (the two grants provide equal support, Z.D.), No. 2019YFA09006700 (L.D.), the U.S. Army Research Office under grant #W911NF-14-1-0490 (L.Y.), National Institutes of Health (L.Y.: 2R01-GM098642, 1RO1AI25604), a David and Lucile Packard Fellowship (L.Y.), Shenzhen Peacock Team Project (KQTD20180413181837372) (Z.D.), National Natural Science Foundation of China No. 32071427 (Z.D.), No. 31971513 (L.D.).

## Author contributions

Z.D. conceived the research, designed and performed experiments, interpreted the results, and wrote the manuscript. X.Y. assisted in performing experiments, data interpretation, and manuscript revisions. F.W. assisted in analyzing 16s sequencing data and manuscript revisions. L.W., K.X., and Q.L. assisted in animal experimental setup, data interpretation, and manuscript revisions. P.L. assisted in the experimental setup during the manuscript revision. J.T. and A.D. assisted in analyzing 16s sequencing data and manuscript revisions. L.D. assisted in research design, experimental setup, data interpretation, and manuscript revisions. X.S. assisted in research design, experimental setup, data interpretation, and manuscript revisions. L.Y. conceived the research, assisted in research design, data interpretation, and manuscript writing.

## Competing interests

The authors declare no competing interests.
