## [Peer Review File · Nature Communications]

REVIEWER COMMENTS

Reviewer #1 (Remarks to the Author):

You et al reported the engineering of living protein materials with an interpenetrating polymer network structure. They encapsulated engineered bacteria within a chitosan scaffold, which can then produce proteins that can be polymerized in situ. They demonstrated that this method can be used to encapsulate proteins that can resist perturbations, and be used to protect gut microbiota in animals from antibiotic-mediated perturbations. The materials presented in this manuscript are an interesting example of the engineered living materials and will be of interests to the researchers in the area of biomaterials. However, the progress demonstrated in this manuscript is largely incremental compared with the previous state-of-the-art of the field of living materials. And the claimed advantages of the engineered living materials versus traditional encapsulation need to be substantiated.

1) The authors claimed the engineering of living materials with an IPN structure. However, the experimental data does not support this major claim. Since the authors ran SDS-PAGE on the crosslinking reaction (Fig. S3) on the proteins at the similar concentration as those produced by the bacteria in the scaffold, this result suggested that the C-mer and T-mer did not form a gel at this particular concentration. How can the proteins at a similar concentration form the second interpenetrating network in the scaffold? It is more likely that the proteins oligomerized within the scaffold and remained trapped there due to their high molecular weight. Therefore, the major claim of the formation of IPN lacks experimental support.

2) The comparison between the so-called IPN materials versus non-IPN is not valid, as there is no secondary network in the non-IPN material. The fact that m-cherry fluorescence remained in the IPN material could be simply due to the physical trapping of the polymerized (but not crosslinked) proteins due to their large MW. In this vein, Fig. S6 does not prove the formation of IPN structure.

3) Fig. 3: the authors claimed that living Bla-IPN materials help recover Bla function after transient or prolonged perturbations. In this experiment, the authors used the Bla activity in the supernatant as a control. A control with the non-IPN material containing Bla would be more suitable. Furthermore, in Fig. S7, the authors showed that all the clavulanic acid treated living Bla-material resumed the activity after another 24 hours culture. To prove that the recovery of the activity is due to the newly produced Bla, a control experiment is needed to prove the simple encapsulation of Bla will not have the same effect.

4) Fig. 2A: T2 shows three distinct bands. What are they? The identity and purity of the produced proteins are questionable.

5) It will be helpful if the authors could briefly explain the formation of the network by chitosan.

Reviewer #2 (Remarks to the Author):

Dai et al provide a very interesting manuscript, in which the authors demonstrate the living reinforcement of cells encapsulated in the natural polymer chitosan. Furthermore, they demonstrate efficacy in the in vitro and in vivo use of enzyme-functionalized materials to breakdown b-lactam antibiotics. The work is innovative and will be of great interest to the materials science and biomedical communities. Especially impressive is the fact that monomers did not need to be purified in the process, making this technique much more versatile to generate ELMs. Several suggestions below, I believe, would improve upon the paper - the most critical of which is a better understanding of the materials properties of the reported composite polymers (i.e. is it really an IPN or is it a gel enhanced with a stiff filler?).

- The major critique in this manuscript is related to the discussion of making living interpenetrating networks (IPNs). The complex nature of the materials can lead to many potentially confounding effects. In non-IPN systems cells are growing and producing rigid 'monomers' which are likely hard nanometer sized filler. So, do the resultant increases in modulus reflect an IPN or just an increase in functional filler (i.e. even if the proteins polymerize but don't form an IPN, the gel modulus would still increase). Likewise, Figure S4 looks identical to me between non-IPN and IPN images. This could be potentially teased out through in vitro experiments with no cells, cells that do not produce any monomers, and differential staining of polymer components.

- Some other minor points: The monomers aren't described well in the text, the reader has to go to the SI to figure out what is being made. A few sentences of description would be helpful. Same for the 'ePop circuit'. This manuscript will likely receive broad readership in the polymer science community and those terms should be more explicitly described. Same for SpyTag and SpyCatcher.

- The reported system is microcapsules. A discussion about application to volumes beyond 1 mL would be quite helpful in evaluating how broadly applicable this technology is and where it may need to go to improve to large-scale materials. Advantages, challenges, etc.

Reviewer #3 (Remarks to the Author):

The manuscript by Dai et al, describes a living material system where E. coli bacteria are encapsulated in chitosan spheres. Within these they can grow, divide and be genetically engineered to occasionally lyse and release proteins. These proteins would normally leave the spheres by diffusion, but here the authors show that by designing them to spontaneously link together by SpyTag/Catcher mediated covalent fusion once outside the cell, they are instead held within the sphere, presumably due to networks of interlinked proteins being established. Using beta lactamase as an example, the authors show that when an enzyme is made as part of this network it has improved performance and recovery from inhibition. As an application, spheres with this enzyme are fed to mice that are simultaneously fed antibiotic and the presence of spheres prevent the antibiotic disrupting the bacteria in the gut.

On the whole this is an interesting and innovative piece of research, which nicely blends medically-relevant materials with synthetic biology. It is a little disappointing that the work is only really demonstrated with one enzyme (bla) as I expect there are many other useful things that can be done using other enzymes, and people who want to adopt this technology would ideally like to have confidence that it works with more than one example. However, that being said I think the has enough reach to show proof-of-concept.

While I'm supportive of publication, I do think there are several places in the manuscript where things need to be clarified or improved. A lot of the presented data is lacking in the sort of details required to make the claims made in the manuscript and I think the authors will need to either provide the extra data or revise their claims. Below are a list of my comments.

Comments

1. The manuscript title, abstract and introduction are written in a way that made me think that the bacteria inside the spheres produce a material component that replenishes and adds to the chitosan material. I later realised that this is not the case, and that no major material component is made by the living cells – they are instead just polymerising their lysed proteins into aggregates that cannot leave the spheres. I think it is important that these sections (title, abstract and introduction) are properly written to not make claims that are not fully-supported. In particular the paper's central claim is that the proteins form an 'interpenetrating polymer network' but I didn't see any direct evidence (e.g images) for the proteins themselves making their own network that goes in and among the much larger chitosan network. I couldn't rule out that the method just leads to aggregates of the protein that are too large to diffuse through the chitosan. This is not the same as a material network.

I also found that the second paragraph of the abstract was not well written.

2. The description and details of the synthetic biology work in this paper are lacking. There is no good description of what the protein networks are made of in the main text and even in the

methods section there is hardly any information. Only after looking at a reference to another paper did I realise that the networks contained elastin proteins, which would be a key part of the network. The authors need to have a figure or diagram to better describe the proteins and arrangements of proteins used in this paper if readers are going to understand how they can do similar work with other proteins in the future.

3. The authors don't justify why the E.coli cells need to 'lyse' to make the protein networks. Already one other synthetic biology group has demonstrated bacteria engineered to make similar SpyTag-SpyCatcher polymers/networks by secreting the proteins (rather than lysis). This group used elastin components and other enzymes too and is quite similar. This paper is not even cited in the manuscript, but should be a key reference - <https://pubs.acs.org/doi/abs/10.1021/acssynbio.6b00292>

4. For Fig 2, the claim is made that the encapsulation provides 10x more monomer than liquid culture, but this is only from indirect measurement of enzymatic activity. Could this not simply be that the enzymatic activity is better in the environment of within the sphere? I would prefer a more direct way to back up this claim.

5. Figure S3 is compared against Figure 2A to say that the polymerisation is better inside the sphere, but I noticed that the two protein gels look very different when side-to-side, both in terms of exposure and in terms of where the authors have cut off the highest weight. This seems a concern to me. Figure S3 is highlighting high MW products (>460 kDa) and saying these don't exist in Figure 2A, but the equivalent gel in Figure 2A is cropped so that these products of this size are not shown. I think to be able to make these claims, the samples should be run on the same gel.

6. The two SI videos seem to be labelled as the opposite of what makes sense. SI-Video 1 showed no RFP leaking, but the text said RFP leaks in this one

7. The SI videos have numbers in the top left (time?) but no indication of the scale of these numbers (minutes, hours?)

8. SI videos show the spheres shrinking initially, with this more pronounced when the RFP leaks out of the sphere. Can the authors comment on this? This seemed relevant but was not mentioned in the paper

9. The SI videos seem to show some cells/debris escaping from the spheres. Can the authors clarify whether E.coli cells leave the spheres in either the normal or IPN versions of the spheres? This would be a major point worth knowing and quantifying for downstream use by others.

10. Figure S4 claims to show polymerised structure dotted with mCherry, but this is very weak evidence. It's not very clear and could simply be a case of finding one part of an image that looks like what the authors hope to see. I don't think this is convincing at all. With only one image for each case it's hard to make a judgement.

11. Also for the SEM images, only one example image is shown which makes it hard to be convinced. The selected images do look quite different, but is this really representative? The images also don't show where the cells are or any evidence for a protein network. I think TEM or environmental SEM images would be more appropriate here to show these.

12. To really show an IPN, I think the authors should make a version with a metal-binding tag (e.g. using His-tagged proteins in their IPN) and then before SEM or eSEM or TEM they can soak the sphere with metal beads (e.g. Ni/Au-beads). These will then be directly visualised within the material

by the electron microscopy, showing whether a network of protein is present, or whether there is simply just protein aggregates from lysed cells.

13. Figure S7a figure legend needs to be rewritten as it doesn't make sense to be talking about 'directions'

14. All the figures using Bla show the data normalised as 'Bla activity'. I would be more comfortable if I could see some of the non-normalised data first. I suspect that the Bla activity of the spheres is very different to that of the free Bla enzyme, and that may help explain many of the observations seen in the figures. Is the bla activity equal in these experiments or is one 10x stronger than the other?

15. Figure 3 makes the claim that Bla is protected by being in the spheres (activity is still seen when inhibitor is added) but could this instead be due to the inhibitor just not diffusing into the spheres very well? The authors should show evidence that clavulanic acid can diffuse into the spheres and block Bla efficiently.

16. On line 186 the authors claim that the spheres protect the E.coli at pH1.0, but Figure S10 shows a 100-fold reduction in CFU. So I don't think they really protect very well.

17. In Figure 4, the results for the capsules containing pure Bla look comparable to the results containing the living Bla IPN capsule. I'm not convinced that there is a statistically significant difference just by looking at the data presented in the figures in panels A and B.

18. I don't understand the timescales in Figure 4. The authors say that the capsules were given before the ampicillin was given. The capsules only stayed in the gut for 4 to 5 hours, but the ampicillin was given for 3 days and the experiment run for 14 days. This doesn't make sense as antibiotic would be given after all the capsules had gone. Can this be clarified?

Below, we provide point-by-point responses to reviewers' comments (*italicized, blue*). The revision in the manuscript is highlighted in blue.

Reviewer #1: You et al reported the engineering of living protein materials with an interpenetrating polymer network structure. They encapsulated engineered bacteria within a chitosan scaffold, which can then produce proteins that can be polymerized in situ. They demonstrated that this method can be used to encapsulate proteins that can resist perturbations, and be used to protect gut microbiota in animals from antibiotic-mediated perturbations. The materials presented in this manuscript are an interesting example of the engineered living materials and will be of interests to the researchers in the area of biomaterials. However, the progress demonstrated in this manuscript is largely incremental compared with the previous state-of-the-art of the field of living materials. And the claimed advantages of the engineered living materials versus traditional encapsulation need to be substantiated.

We thank the reviewer for evaluating our manuscript. His/her comments reveal multiple aspects that we should further clarify, including the conceptual novelty, approach, scope, and implications.

As the reviewer mentioned, a few recent examples have demonstrated the use of engineered bacteria to fabricate materials. In these systems, the living cells are used during the material fabrication process¹²⁻¹⁵, instead of being used as the integral part of the final material. That is, the functions of the resulting materials do not critically depend on having the living cells^{13,16-19}.

In this work, we demonstrate a fundamental new approach to fabricate material that is living, functional and biocompatible. The fabrication is driven by the engineered bacteria encapsulated in a polymeric microcapsule, which serves as the initial scaffold. The bacteria grow and undergo programmed lysis in a density-dependent manner, releasing protein monomers decorated with reactive tags. Those protein monomers polymerize with each other to form the second polymeric component that is interlaced with the initial polymeric scaffold.

In comparison with conventional methods (especially those methods used in traditional encapsulation method for enzyme immobilization), our strategy does not require the labor-intensive processes of preparing the purified components^{20,21}. The protein component in the living material is assembled entirely by the monomers produced and released by the engineered bacteria. Due to the circuit design, the living functional material is resilient (in response to perturbations) and robust (in terms of maintaining the programmed functions). In particular, it represents the first example to use living cells to create active materials, **whose function also requires programmed behavior of the cells**. Moreover, to our knowledge, this is the first example where bacterially fabricated living materials are used in a setting with direct practical relevance – the protection of the gut microbiota against unintended perturbations by antibiotics. For this application, the unique features of the living materials (i.e., biocompatibility, resilience, and robustness) are critical for the target application.

1) The authors claimed the engineering of living materials with an IPN structure. However, the experimental data does not support this major claim. Since the authors ran SDS-PAGE on the crosslinking reaction (Fig. S3) on the proteins at the similar concentration as those produced by the bacteria in the scaffold, this result suggested that the C-mer and T-mer did not form a gel at this particular concentration. How can the proteins at a similar concentration form the second interpenetrating network in the scaffold? It is more likely that the proteins oligomerized within the scaffold and remained trapped there due to their high molecular weight. Therefore, the major claim of the formation of IPN lacks experimental support.

We thank the reviewer for this question. Indeed, we need to better clarify our terminology and the conceptual progress of our work.

Interpenetrating polymer network (IPN) is a board concept. It is generally defined as a blend of two or more polymers in a network with at least one of the systems synthesized in the presence of another. In the literature, IPN is classified in two categories¹⁻⁵. Based on the structure, IPN is classified into semi- and full

IPN, depending on whether one or both of the respective components are fully crosslinked⁶⁻⁸. If both components are fully crosslinked, the system is called full IPN. If one component is fully crosslinked and the other component is in polymer state but not crosslinked, the system is called semi-IPN. Based on method of synthesis, IPN is classified into sequential or simultaneous IPN. In sequential IPN, the first crosslinked polymer network is swollen by the monomer of the second polymer that is polymerized and/or cross-linked afterwards^{9,10}. While in simultaneous IPN, both components are polymerized concurrently¹¹.

In our work, we demonstrate the engineering of living materials consisting of programed bacteria and non-living components. The fabrication process is driven by the engineered bacteria encapsulated in a polymeric microcapsule (crosslinked), which serves as the initial scaffold (Figure S2). The bacteria grow and undergo programmed lysis in a density-dependent manner, releasing protein monomers decorated with reactive tags (Figure 2A). Those protein monomers polymerize with each other to form the second polymeric component that is interlaced with the initial crosslinked scaffold (Figure 2, S4 and S7). **Our results demonstrate that the characteristics of resulting materials are fully consistent with the definition of a semi-IPN. Therefore, our definition of the system (living IPN capsules) is correct.**

Based on our fabrication method, the second polymeric component (protein component) forms in the presence of the first polymer network; thus, it is a sequential IPN. A unique feature of our system is that the second set of monomers were produced and released autonomously by the living bacteria entrapped in the first network. This aspect represents a conceptually novel approach of IPN fabrication. **In light of the reviewers' comment and to avoid potential confusion, we have revised our text and changed the terminology to semi-IPN (sIPN capsules).**

Last but not least, **the core concept and innovation of this work is to develop living functional materials** by integrating engineered bacteria and non-living components. Due to the use of engineered bacteria capable of growth and regulated lysis, the living functional material is **resilient to transient perturbations** and robust in maintaining the programmed functions (Figure 3, S9 and S10). The living material formed, has **an improved mechanical properties** (Figure S8). The key novelty and advantage of the system, no matter how we name the system, remain unchanged based on our data. We will further clarify this point in the revised manuscript.

2) The comparison between the so-called IPN materials versus non-IPN is not valid, as there is no secondary network in the non-IPN material. The fact that m-cherry fluorescence remained in the IPN material could be simply due to the physical trapping of the polymerized (but not crosslinked) proteins due to their large MW. In this vein, Fig. S6 does not prove the formation of IPN structure.

We thank the reviewer for this comment. We did not claim the released protein formed a fully network. Instead, we used “polymerized protein” throughout the manuscript. In light of the reviewer’s comments, we have revised the manuscript to better reflect this aspect. We hope the term semi-IPN can eliminate the confusion.

To further examine the structure of the polymerized protein, we used a differential staining method to stain the polymerized protein with gold nanoparticles. Briefly, the protein monomer was modified with a metal-binding tag (His-tagged). Both living sIPN and non-IPN capsules were first dehydrated, embedded and sliced into thin sections. The sections were placed on a copper grid and stained with Ni-NTA-AuNPs (5 nm) for TEM examination. Our results showed the polymerized protein structure in living sIPN capsules as indicated by the densely alignment of the gold nanoparticles. In comparison, there is only sparse and scattered distribution of gold nanoparticles in the non-IPN capsules.

The improvement in the mechanical property by semi-IPN formation has been reported by multiple previous studies^{22,23}. For example, Fernandez-Gutierrez *et al* synthesized semi-interpenetrating polymer

networks (semi-IPN) by free radical polymerization of N-isopropylacrylamide, in the presence of chitosan²⁴. They observed that the elastic modulus increased as function of the percentage of chitosan added into semi-IPN. In another example, Han *et al* reported semi-interpenetrating polymeric networks: PHEMA/chitosan or P(HEMA-co-SMA)/chitosan by crosslinking poly(2-hydroxyethyl methacrylate) (PHEMA) or poly(2-hydroxyethyl methacrylate-co-sodium methacrylate) [P(HEMA-co-SMA)], in the presence of chitosan with different molecular weight²⁵. The tensile properties and compressive modulus of resultant semi-IPN increased as the molecular weight of chitosan increased.

We further measured the mechanical strength of **capsules containing bacteria not producing any monomers** (Figure S8). We encapsulated the cells (MC4100Z1(ePop), same initial density with living sIPN and non-IPN capsules), cultured with the same amount of nutrients, and measured the mechanical strength of the resultant capsules. Our results showed that the storage modulus of capsules containing bacteria not producing any monomers increased gradually with time. Especially, this increase in storage modulus is comparable with the non-IPN capsules. Therefore, we reasoned that the increase of the storage modulus of both capsules containing bacteria not producing any monomers and non-IPN capsules (~2 times compared with the original modulus) is possibly due to the increase of the cell mass. The increase (~4 times compared with the original modulus) in the mechanical strength of living sIPN capsules is likely due the presence of the polymerized protein and the formation of semi-IPN.

We have further clarified this point in the revised manuscript.

3) Fig. 3: the authors claimed that living Bla-IPN materials help recover Bla function after transient or prolonged perturbations. In this experiment, the authors used the Bla activity in the supernatant as a control. A control with the non-IPN material containing Bla would be more suitable.

We thank the reviewer for this constructive comment. In light of the reviewer's comment, we further evaluated the activity of living non-IPN capsules producing Bla under transient or prolonged perturbations (Figure S10A and B). Our results showed that when clavulanic acid was supplemented for 24 hours (co-cultured with 0.1, 0.5 or 1 $\mu\text{g}/\text{mL}$ clavulanic acid overnight, constant perturbation), non-IPN capsules maintained ~10%-45% of the enzymatic activity; in contrast, the living sIPN capsules maintained ~65%-85% of the enzymatic activity. When clavulanic acid was supplemented for 10 mins (transient perturbation), the living non-IPN capsules producing Bla remained ~ 37%, 29% or zero activity when treated by 0.1, 0.5 or 1 $\mu\text{g}/\text{mL}$ clavulanic acid. After another 24 hours culture, the clavulanic acid treated living non-IPN capsules resumed the 87%, 53% and 53% activity. In contrast, living sIPN capsules could partially stabilize the Bla function and maintained the ~ 65%, 45% or 10% of the enzymatic activity when treated by 0.1, 0.5 or 1 $\mu\text{g}/\text{mL}$ clavulanic acid. All the clavulanic acid treated living sIPN capsules resumed the activity after another 24 hours culture (Figure S9B). Our results confirmed a more pronounced stabilization effect by living sIPN material under both transient and prolonged perturbation.

We have revised the main text and figures accordingly.

Furthermore, in Fig. S7, the authors showed that all the clavulanic acid treated living Bla-material resumed the activity after another 24 hours culture. To prove that the recovery of the activity is due to the newly produced Bla, a control experiment is needed to prove the simple encapsulation of Bla will not have the same effect.

We thank the reviewer for this constructive comment. In light of the reviewer's comment, we evaluated the recovery capability of capsules encapsulating pure Bla after transient perturbation (Figure S10C). When clavulanic acid was supplemented for 10 mins (transient perturbation), capsules containing pure Bla remained ~ 24%, 9% or 3% of original activity when treated by 0.1, 0.5 or 1 $\mu\text{g}/\text{mL}$ clavulanic acid. After another 24 hours culture, there was no recovery in the activity.

4) Fig. 2A: T2 shows three distinct bands. What are they? The identity and purity of the produced proteins are questionable.

We thank the reviewer for this question. T₂ is His-SpyTag-ELPs-mCherry-ELPs-SpyTag (SI-method). The presence of the three distinct bands is due to the fusion of mcherry protein. The mCherry protein is known to be unstable and result multiple bands when run on SDS-PAGE. Multiple lab and bio-company reported this phenomena in recombinant mcherry protein (<https://encorbio.com/product/prot-mcherry/>) or recombinant mcherry fused protein (<https://www.ptglab.com/products/mCherry-Fusion-Protein-Ag25320.htm>). One possible explanation is that SDS treatment and boiling before PAGE hydrolyzes and splits the protein in half²⁶.

5) It will be helpful if the authors could briefly explain the formation of the network by chitosan.

We thank the reviewer for this constructive comment. We have elaborated more in the text to better explain the formation and property of chitosan network. Briefly, TPP (tripolyphosphate) is a non-toxic polyanion negative-charged phosphate groups of TPP interact with positive-charged amine groups in chitosan and form ionic crosslinked networks^{27,28}.

We thank the reviewer again for evaluating our manuscript and providing instructive comments.

Reviewer #2: Dai et al provide a very interesting manuscript, in which the authors demonstrate the living reinforcement of cells encapsulated in the natural polymer chitosan. Furthermore, they demonstrate efficacy in the in vitro and in vivo use of enzyme-functionalized materials to breakdown β -lactam antibiotics. The work is innovative and will be of great interest to the materials science and biomedical communities. Especially impressive is the fact that monomers did not need to be purified in the process, making this technique much more versatile to generate ELMs. Several suggestions below, I believe, would improve upon the paper - the most critical of which is a better understanding of the materials properties of the reported composite polymers (i.e. is it really an IPN or is it a gel enhanced with a stiff filler?).

We thank the reviewer for recognizing the novelty and significance of our study and for providing insightful comments. In light of the reviewer's comments, we have thoroughly revised the manuscript and provided additional experimental data to address the reviewer's questions.

1. The major critique in this manuscript is related to the discussion of making living interpenetrating networks (IPNs). The complex nature of the materials can lead to many potentially confounding effects. In non-IPN systems cells are growing and producing rigid 'monomers' which are likely hard nanometer sized filler. So, do the resultant increases in modulus reflect an IPN or just an increase in functional filler (i.e. even if the proteins polymerize but don't form an IPN, the gel modulus would still increase).

The improvement in the mechanical property by semi-IPN formation has been reported by multiple previous studies^{22,23}. For example, Fernandez-Gutierrez *et al* synthesized semi-interpenetrating polymer networks (semi-IPN) by free radical polymerization of N-isopropylacrylamide, in the presence of chitosan²⁴. They observed that the elastic modulus increases as function of the percentage of chitosan added into semi-IPN. In another example, Han *et al* reported semi-interpenetrating polymeric network: PHEMA/chitosan or P(HEMA-co-SMA)/chitosan by crosslinking poly(2-hydroxyethyl methacrylate) (PHEMA) or poly(2-hydroxyethyl methacrylate-co-sodium methacrylate) [P(HEMA-co-SMA)], in the presence of chitosan with different molecular weight²⁵. The tensile properties and compressive modulus of resultant semi-IPN increased as the molecular weight of chitosan increased.

In light of reviewer's advice, we further measured the mechanical strength of **capsules containing bacteria not producing any monomers** (Figure S8). We encapsulated the cells (MC4100Z1(ePop), same initial density with living sIPN and non-IPN capsules), cultured with the same amount of nutrients, and measured the mechanical strength of the resultant capsules. Our results showed that the storage modulus of capsules containing bacteria not producing any monomers increased gradually with time. Especially, this increase in storage modulus is comparable with the non-IPN capsules. Therefore, we reasoned that the increase of the storage modulus of both capsules containing bacteria not producing any monomers and non-IPN capsules (~2 times compared with the original modulus) is possibly due to the increase of the cell mass. The increase (~4 times compared with the original modulus) in the mechanical strength of living sIPN capsules is likely due the presence of the polymerized protein and the formation of semi-IPN.

Likewise, Figure S4 looks identical to me between non-IPN and IPN images. This could be potentially teased out through in vitro experiments with no cells, cells that do not produce any monomers, and differential staining of polymer components.

As suggested by the reviewer, we have further examined the internal structure of the polymerized protein using a differential staining method (Figure S7). Briefly, the protein monomer was modified with a metal-binding tag (His-tagged). Both living sIPN and non-IPN capsules were dehydrated, embedded and sliced into thin sections. The sections were placed on a copper grid and stained with Ni-NTA-AuNPs (5 nm) for TEM examination. Our results showed the polymerized protein structure in living sIPN capsules as indicated by the densely alignment of the gold nanoparticles. In comparison, there is only sparse and scattered distribution of gold nanoparticles in the non-IPN capsules.

- 2. Some other minor points: The monomers aren't described well in the text, the reader has to go to the SI to figure out what is being made. A few sentences of description would be helpful. Same for the 'ePop circuit'. This manuscript will likely receive broad readership in the polymer science community and those terms should be more explicitly described. Same for SpyTag and SpyCatcher.*

We thank the reviewer for this constructive comment. We have elaborated more about the monomers, the Spytag-Spycatcher pair and ePop circuit in our text for better understanding, as highlighted by the blue color. We have also revised figure 2A and supplemented a figure (Figure S1) to describe the protein monomers.

The backbone of the monomers is elastin-like polypeptides (ELPs). These ELPs were fused with either multiple SpyCatcher or SpyTag sequences, so they can polymerized by covalent bonding. Elastin is a connective tissue protein that imparts both strength and flexibility to the extracellular matrix. Owing to their high solubility, high-yield expression in *Escherichia coli*, and tunable lower critical solution temperatures (LCSTs), ELPs serve as versatile model systems for biomaterials development. ELPs compose tandem pentapeptides of the form (VPGXG)_n. The ELPs used in this work compose a mixture of (VPGEG) and (VPGVG) motifs to raise the LCST above 37 °C and enable the fully swollen chains at culture temperature.

Howarth and coworkers designed a pair of reactive protein partners (SpyTag and SpyCatcher) by splitting the second immunoglobulinlike collagen adhesin domain (CnaB2) of the fibronectin-binding protein (FbaB) of *Streptococcus pyogenes*. SpyCatcher and SpyTag can spontaneously react to reconstitute the intact folded CnaB2 domain. Under mild conditions, the SpyCatcher-SpyTag chemistry proceeds efficiently at temperatures ranging from 4 to 37 °C and does not require additional chemical reagents or catalysts.

We have also elaborated more on the logic of the ePop circuit in the revised manuscript. Briefly, high cell densities lead to an increased plasmid copy number and greater E protein expression (a toxic protein interferes the cell wall synthesis), which then leads to lysis of a subpopulation of the bacteria²⁹. From our previous study (Marguet et al, PLoS ONE 2010, Figure S1F), the majority of cells (~90% by CFU counts) would lyse during each cycle in the liquid media, which exceeds what OD values reveal³⁰.

- 3. The reported system is microcapsules. A discussion about application to volumes beyond 1 mL would be quite helpful in evaluating how broadly applicable this technology is and where it may need to go to improve to large-scale materials. Advantages, challenges, etc.*

We thank the reviewer for this constructive comment. Our strategy could be readily extended to build bulk living material due to its modularity. The engineered gene circuit, the bacterial strain, the functional protein, the crosslinking chemistry and encapsulating material can be separately optimized and then integrated. For example, we can mix alginate with engineered bacteria, and crosslink the system with calcium ions to form bulk gel^{7,31}. The bacteria inside the bulk gel grow and lysis to release the monomers, which can react inside the alginate to form the polymerized protein. We have elaborated this part in the discussion.

We thank the reviewer again for evaluating our manuscript and providing instructive comments.

Reviewer #3: The manuscript by Dai et al, describes a living material system where E. coli bacteria are encapsulated in chitosan spheres. Within these they can grow, divide and be genetically engineered to occasionally lyse and release proteins. These proteins would normally leave the spheres by diffusion, but here the authors show that by designing them to spontaneously link together by SpyTag/Catcher mediated covalent fusion once outside the cell, they are instead held within the sphere, presumably due to networks of interlinked proteins being established. Using beta lactamase as an example, the authors show that when an enzyme is made as part of this network it has improved performance and recovery from inhibition. As an application, spheres with this enzyme are fed to mice that are simultaneously fed antibiotic and the presence of spheres prevent the antibiotic disrupting the bacteria in the gut.

On the whole this is an interesting and innovative piece of research, which nicely blends medically-relevant materials with synthetic biology. It is a little disappointing that the work is only really demonstrated with one enzyme (bla) as I expect there are many other useful things that can be done using other enzymes, and people who want to adopt this technology would ideally like to have confidence that it works with more than one example. However, that being said I think the has enough reach to show proof-of-concept.

While I'm supportive of publication, I do think there are several places in the manuscript where things need to be clarified or improved. A lot of the presented data is lacking in the sort of details required to make the claims made in the manuscript and I think the authors will need to either provide the extra data or revise their claims. Below are a list of my comments.

We thank the reviewer for recognizing the novelty and significance of our study and for providing insightful comments. In light of the reviewer's comments, we have thoroughly revised the manuscript and provide additional experimental data to address the reviewer's question.

1. The manuscript title, abstract and introduction are written in a way that made me think that the bacteria inside the spheres produce a material component that replenishes and adds to the chitosan material. I later realised that this is not the case, and that no major material component is made by the living cells – they are instead just polymerising their lysed proteins into aggregates that cannot leave the spheres. I think it is important that these sections (title, abstract and introduction) are properly written to not make claims that are not fully-supported. In particular the paper's central claim is that the proteins form an 'interpenetrating polymer network' but I didn't see any direct evidence (e.g images) for the proteins themselves making their own network that goes in and among the much larger chitosan network. I couldn't rule out that the method just leads to aggregates of the protein that are too large to diffuse through the chitosan. This is not the same as a material network.

I also found that the second paragraph of the abstract was not well written.

We thank the reviewer for this very helpful comment. Indeed, we need to better clarify our terminology.

Interpenetrating polymer network (IPN) is a board concept. It is generally defined as a blend of two or more polymers in a network with at least one of the systems synthesized in the presence of another. According to classical textbooks and references, two categorizing schemes are used to describe IPN¹⁻⁵. Based on the structure, IPN are classified into semi- and full IPN, depending on whether one or both of the respective components are crosslinked⁶⁻⁸. Based on method of synthesis, IPN is classified into sequential or simultaneous IPN. In sequential IPN, the first crosslinked polymer network is swollen by the monomer of the second polymer that is polymerized and/or cross-linked afterwards^{9,10}. While in simultaneous IPN, both components are polymerized concurrently¹¹.

In our work, we demonstrate the engineering of living materials consisting of programed bacteria and non-living components. The fabrication process is driven by the engineered bacteria encapsulated in a polymeric microcapsule (crosslinked), which serves as the initial scaffold (Figure S2). The bacteria grow and undergo programmed lysis in a density-dependent manner, releasing protein monomers decorated with reactive tags (Figure 2A). Those protein monomers polymerize with each other to form the second polymeric component that is interlaced with the initial crosslinked scaffold (Figure 2, S4 and S7). **Our results fully support these characteristics describe above, and these characteristics meet the requirements defined by a semi-IPN.**

Based on our fabrication method, the second polymeric component (protein component) forms in the presence of the first polymer network, more like a sequential IPN. A fundamental difference is that, in our system, the second set of monomers were produced and released autonomously by the living bacteria entrapped in the first network. The sequential IPN also belongs to the general IPN category. **To avoid further confusion, we have revised whole text and changed the terminology to semi-IPN (sIPN capsules).**

We have revised the second paragraph of the introduction as the reviewer suggested.

2. The description and details of the synthetic biology work in this paper are lacking. There is no good description of what the protein networks are made of in the main text and even in the methods section there is hardly any information. Only after looking at a reference to another paper did I realise that the networks contained elastin proteins, which would be a key part of the network. The authors need to have a figure or diagram to better describe the proteins and arrangements of proteins used in this paper if readers are going to understand how they can do similar work with other proteins in the future.

We thank the reviewer for this constructive comment. We have elaborated more about the monomers, the Spytag-Spycatcher pair and ePop circuit in our text for better understanding, as highlighted by the blue color. We have also supplemented a figure (Figure S1) to describe the protein monomers.

The backbone of the monomers is elastin-like polypeptides (ELPs). These ELPs were fused with either multiple SpyCatcher or SpyTag sequences, so they can polymerized by covalent bonding. Elastin is a connective tissue protein that imparts both strength and flexibility to the extracellular matrix. Owing to their high solubility, high-yield expression in *Escherichia coli*, and tunable lower critical solution temperatures (LCSTs), ELPs serve as versatile model systems for biomaterials development. ELPs compose tandem pentapeptides of the form (VPGXG)_n. The ELPs used in this work comprise a mixture of (VPGEG) and (VPGVG) motifs to raise the LCST above 37 °C and enable the fully swollen chains at culture temperature.

Howarth and coworkers designed a pair of reactive protein partners (SpyTag and SpyCatcher) by splitting the second immunoglobulinlike collagen adhesin domain (CnaB2) of the fibronectin-binding protein (FbaB) of *Streptococcus pyogenes*. SpyCatcher and SpyTag can spontaneously react to reconstitute the intact folded CnaB2 domain. Under mild conditions, the SpyCatcher-SpyTag chemistry proceeds efficiently at temperatures ranging from 4 to 37 °C and does not require additional chemical reagents or catalysts.

We have also elaborated more on the logic of the ePop circuit in the revised manuscript. Briefly, high cell densities lead to an increased plasmid copy number and greater E protein expression (a toxic protein interferes the cell wall synthesis), which then leads to lysis of a subpopulation of the bacteria²⁹. From our previous study (Marguet et al, PLoS ONE 2010, Figure S1F), the majority of cells (~90% by CFU counts) would lyse during each cycle in the liquid media, which exceeds what OD values reveal³⁰.

3. The authors don't justify why the E.coli cells need to 'lyse' to make the protein networks. Already one other synthetic biology group has demonstrated bacteria engineered to make similar SpyTag-SpyCatcher polymers/networks by secreting the proteins (rather than lysis). This group used elastin components and other enzymes too and is quite similar. This paper is not even cited in the manuscript, but should be a key reference - <https://pubs.acs.org/doi/abs/10.1021/acssynbio.6b00292>

We thank the reviewer for this constructive comment. The reviewer is absolutely correct that lysis is not the only mechanism to release proteins. Depending on the application context, secretion can be an effective alternative³². Yet, the diverse secretion mechanism of bacteria can make the selection of the appropriate secretion pathway for each recombinant protein complicated and time consuming³³. Besides, the efficiency of secretion is often limited in bacterial hosts and specific proteins to be secreted. Engineered bacteria capable of programmed autonomous lysis do not suffer from these limitations: any protein that can be expressed at sufficiently high levels can be released, without the need for extensive optimization.

The work mentioned by the reviewer is indeed a very relevant work. We have cited the work and further elaborated the rationale of using engineered bacteria capable of programmed, autonomous lysis.

4. For Fig 2, the claim is made that the encapsulation provides 10x more monomer than liquid culture, but this is only from indirect measurement of enzymatic activity. Could this not simply be that the enzymatic activity is better in the environment of within the sphere? I would prefer a more direct way to back up this claim.

We thank the reviewer for this constructive comment. 10x monomer concentration is an estimation based on the monomers trapped in the capsules (~400 μ L, assuming all the monomers were kept in the capsules compared with releasing to the supernatant) than released into the supernatant (4 mL). As suggested by the reviewer, we released the monomer from capsules to directly quantify the enzymatic activity. We first harvested the capsules, released the monomers (see methods) and directly measured the enzymatic activity in the supernatant. We then back-calculated the concentration inside the capsules. Our results show that the concentration measured directly is comparable with the value measured indirectly. We have updated the figure S3B.

5. Figure S3 is compared against Figure 2A to say that the polymerization is better inside the sphere, but I noticed that the two protein gels look very different when side-to-side, both in terms of exposure and in terms of where the authors have cut off the highest weight. This seems a concern to me. Figure S3 is highlighting high MW products (>460 kDa) and saying these don't exist in Figure 2A, but the equivalent gel in Figure 2A is cropped so that these products of this size are not shown. I think to be able to make these claims, the samples should be run on the same gel.

We thank the reviewer for the careful examination and suggestion. Data related with figure 2A was generated in Duke University using a pre-casting gel (<https://www.bio-rad.com/en-sg/product/mini-protean-tgx-stain-free-precast-gels?ID=N3GRU1E8Z>). The uncropped gel is shown as below. We cropped the gel for better presentation and did not exclude products with high molecular weight.

Data related with figure S3 (previous manuscript) was generated in SIAT, CAS using a self-made PAGE gel. In light of the reviewer's comment, we ran both reactions on the same gel. We used 6% gel to separate the large molecular weight product. Our data (figure S4 in the current manuscript) confirmed the previous claims: there is high molecular weight product (>460kDa) present in the at the monomer concentration mimicking the capsule interior. However, the gel cannot separate the protein (especially T₃) below 40kDa. Therefore, we kept the original figure 2A and replaced the S4 with new data according to the reviewer's suggestion.

6. The two SI videos seem to be labelled as the opposite of what makes sense. SI-Video 1 showed no RFP leaking, but the text said RFP leaks in this one

We thank the reviewer for very careful examination. We indeed mislabeled these two videos in the main text. We have revised the main text accordingly.

7. The SI videos have numbers in the top left (time?) but no indication of the scale of these numbers (minutes, hours?)

We thank the reviewer for the careful examination. The number in the video indicates the time and unit is hours in both videos. We have revised both videos accordingly.

8. SI videos show the spheres shrinking initially, with this more pronounced when the RFP leaks out of the sphere. Can the authors comment on this? This seemed relevant but was not mentioned in the paper.

We thank the reviewer for the suggestion.

The shrinking of the capsules was mostly due to the cell growth. It is well established that chitosan capsules can response to pH and ionic strength^{27,28,34}. During growth, the bacterial culture can change the growth environment in multiple aspects, including the pH and ionic strength³⁵⁻³⁷. These changes collectively cause shrinking of chitosan capsules, which was reported by our recent study²⁹.

As the reviewer noticed, this phenomenon was more pronounced in the non-IPN capsules but not in the living sIPN-capsules. One possible reason is that the polymerized protein component (hydrophilic ELPs) contributed to the reduced shrinking. We have further elaborated this point in the revised manuscript.

9. The SI videos seem to show some cells/debris escaping from the spheres. Can the authors clarify whether E.coli cells leave the spheres in either the normal or IPN versions of the spheres? This would be a major point worth knowing and quantifying for downstream use by others.

We thank the reviewer for the careful examination and suggestion. In terms of reviewer's comments, we have further quantified the escape rates of bacteria from living non-IPN and sIPN capsules (number of the cells in the capsules/total cell number in capsules). After 24 hours culture, the escape rates for living non-IPN and sIPN were ~0.3% and 0.2%, respectively. The escape rates can be modulated by a surface coating technique. We first coated the capsules with a layer of alginate, followed by another layer of chitosan. After this layer-by-layer modification, the escape rate of the cells in living sIPN capsules decreased to zero, after 24 hours of culturing. We have further elaborated this part in the revised text.

10. Figure S4 claims to show polymerized structure dotted with mCherry, but this is very weak evidence. It's not very clear and could simply be a case of finding one part of an image that looks like what the authors hope to see. I don't think this is convincing at all. With only one image for each case it's hard to make a judgment..

We thank the reviewer for the constructive comments and suggestions. We have supplemented new data and revised our claims accordingly (Figure S5). Compared with the non-IPN capsules (MC(T₂-mCherry) + MC (T₃)), the frozen sections of living sIPN capsules (carrying MC(T₂-mCherry) + MC (C₃)) show overall stronger mCherry signal, which is consistent with what we observed under the fluorescence microscopy (Figure 2B, SI-Video-1 and 2).

11. Also for the SEM images, only one example image is shown which makes it hard to be convinced. The selected images do look quite different, but is this really representative? The images also don't show where the cells are or any evidence for a protein network. I think TEM or environmental SEM images would be more appropriate here to show these.

We thank the reviewer for the constructive comments and suggestions. The images present are indeed representative. We have supplemented more SEM data to show that the internal structure of living sIPN is more porous (Figure S6). In light of reviewer's comments, we have further used TEM to differential stain the polymerized protein structure, as detailed below.

12. To really show an IPN, I think the authors should make a version with a metal-binding tag (e.g. using His-tagged proteins in their IPN) and then before SEM or eSEM or TEM they can soak the sphere with metal beads (e.g. Ni/ Au-beads). These

will then be directly visualised within the material by the electron microscopy, showing whether a network of protein is present, or whether there is simply just protein aggregates from lysed cells.

As suggested by the reviewer, we have further examined the structure of the polymerized protein using a differential staining method (Figure S7). Briefly, the protein monomer was modified with a metal-binding tag (His-tagged). The living non-IPN and sIPN capsules were dehydrated, embedded and sliced into thin sections. The sections were placed on a copper grid and stained with Ni-NTA-AuNPs (5 nm) for TEM examination. Our results showed the polymerized protein structure in living sIPN capsules as indicated by the densely alignment of the gold nanoparticles. In comparison, there is only sparse and scattered distribution of gold nanoparticles in the non-IPN capsules.

13. Figure S7a figure legend needs to be rewritten as it doesn't make sense to be talking about 'directions'

We thank the reviewer for this constructive comment. We have revised the figure legend accordingly.

14. All the figures using Bla show the data normalised as 'Bla activity'. I would be more comfortable if I could see some of the non-normalised data first. I suspect that the Bla activity of the spheres is very different to that of the free Bla enzyme, and that may help explain many of the observations seen in the figures. Is the bla activity equal in these experiments or is one 10x stronger than the other?

We thank the reviewer for this constructive comment.

We would like to first explain how we measured the Bla activity. In our experiments, the Bla activity was quantified by nitrocefin (abcam) assay. Briefly, the conversion of the nitrocefin by Bla into a colored product (detectable at OD = 490) can be described by the Michaelis-Menten kinetics: $v = \lim_{t \rightarrow 0} \frac{dA}{dt} = \frac{kE_0[S]}{K+[S]}$, where v is the initial rate of increase in the absorbance (A), k is the catalytic constant, K is the Michaelis-Menten constant, E_0 is the concentration of Bla, and $[S]$ is the concentration of the substrate (nitrocefin). When the substrate is saturating ($[S] \gg K$), $v = kE_0$, thus v provides a direct measurement of the enzyme activity. Experimentally, v is determined by taking the time derivative of A in the initial time window, when A increases linearly with time.

In light of the reviewer's comments, we have supplemented the Bla activity of living sIPN, non-IPN and supernatant before normalization in the SI (Table S1).

15. Figure 3 makes the claim that Bla is protected by being in the spheres (activity is still seen when inhibitor is added) but could this instead be due to the inhibitor just not diffusing into the spheres very well? The authors should show evidence that clavulanic acid can diffuse into the spheres and block Bla efficiently.

We thank the reviewer for this constructive comment. In light of the reviewer's comments, we have supplemented a control experiments, in which clavulanic acid was added to the capsules containing the purified Bla. When clavulanic acid was supplemented for 10 mins (transient perturbation), capsules containing pure Bla remained ~ 24%, 9% or 3% of original activity when treated by 0.1, 0.5 or 1 $\mu\text{g}/\text{mL}$ clavulanic acid. We have further clarified this point in the revised main text.

16. On line 186 the authors claim that the spheres protect the E.coli at pH1.0, but Figure S10 shows a 100-fold reduction in CFU. So I don't think they really protect very well.

We thank the reviewer for this question. Indeed, the number of viable encapsulated bacteria decreased from $\sim 10^8$ to 10^6 CFU/ mL after being exposed to a highly acidic environment (pH = 1) for 1hr. However, in the liquid culture, the number of viable bacteria decreased from $\sim 10^9$ to 10^3 CFU/ mL after the same treatment. Therefore, the capsules did protect the cells compared with the un-encapsulated cells.

We further tested if the treated cells could re-grow and function properly by culturing the cells (protected or unprotected) in the medium for another 24 hours. Our results showed that the Bla activity produced and released by the protected cells (treated in the acidic environment for 2 hours) was ~90% of the activity under the untreated condition. These results indicate maintenance of the living-synthesis capability by the encapsulated bacteria (Figure 3C). We have further clarified this point in the revised main text.

17. In Figure 4, the results for the capsules containing pure Bla look comparable to the results containing the living Bla IPN capsule. I'm not convinced that there is a statistically significant difference just by looking at the data presented in the figures in panels A and B.

In our original manuscript, we presented data on 20 mice, with 5 mice per group. Our data, while appearing to be noisy, indeed showed statistically significant protection by the living Bla sIPN capsules on Day 4 compared with no capsules group and living non-IPN group ($p < 0.05$).

In light of the reviewer's comment, we repeated the experiment with 10 mice per group (See below for the updated figure 4 and statistical analysis for UniFrac and Table S2). We compared the results from pairs of different conditions by using the two-sided Student's two-sample t-test (assuming unequal variances). These new results further confirmed the conclusion from our original submission: shortly after treatment (Day 4), the living sIPN capsules provided statistically significant protection ($p < 0.05$), compared with each of other three groups. The difference between different groups decreased over time: the microbiome composition was able to recover after the transient antibiotic treatment (with or without protection).

Updated figure 4. Living Bla IPN capsules minimized antibiotic-mediated perturbations of the gut microbiome in mice.

A: Living Bla IPN capsules provided the most potent protection of the gut microbiome from antibiotic-mediated perturbations (*Top: experimental setup; bottom: experimental data*). 40 mice were separated into 4 groups and kept in the facility for one week to stabilize the gut microbiome. Fecal samples were collected on day 0 and used as reference point to evaluate the disturbance to the microbiome. The treatment was given from day 1 to 3. Living Bla IPN capsules, living non-IPN capsules producing Bla or pure Bla capsules were administered to groups 2-4, respectively. After one hour, 5 mg ampicillin (100 μ L) was injected into each mouse through the tail vein for all four groups. Fecal samples from 40 mice were collected on days 0, 4, 7, 14 for DNA quantification and 16S rDNA sequencing. The left panel shows the DNA content (ng DNA per mg fecal matter). Each dot represents an individual mouse, and the column represents the average value ($n=10$). The right panel shows the weighted UniFrac distance between the gut microbiome at Day 4, 7, 14 and the reference point (day 0) as analyzed by QIIME2 (see Methods). The weighted UniFrac distance takes into account both

phylogenetic lineages and relative abundance of two samples and is commonly used as the distance metric for microbial communities. The line represents the average value (n=10). On day 4, the living IPN capsules exhibited the most potent protection in comparison to the other three groups, in terms of maintaining the total DNA content and microbiome composition (** indicates $p < 0.01$, , determined by the student's t-test). The statistical analysis of the weighted UniFrac was shown in Table S2.

B: The relative abundances of major phyla for mice gut microbiota pretreated with different capsules or not pretreated before antibiotic injection. Each row represents a different treatment group. Each column in the panel corresponds to an individual mouse. The x-axis represents the numbering of the mice, and y-axis represents the relative abundance of different phyla. Typically, in the absence of perturbations, Bacteroidetes (blue) and Firmicutes (orange) dominate the mice gut microbiome. In the living Bla IPN group, these two phyla remained dominant and the microbiome composition remained stable (Figure S14). In the absence of pre-treatments, the relative abundance of these major phyla was drastically altered following antibiotics injection.

Table S2: Parametric P-values of the UniFrac distance (updated) calculated using the two-sided Student's two-sample t-test. We compared the results in Figure 4A (UniFrac) from pairs of different conditions of living IPN capsules and other three groups by using the two-sided Student's two-sample t-test (assuming unequal variances). The results show that shortly after treatment (Day 4), the living IPN capsules provided statistically significant protection ($p < 0.05$), compared with each of other three groups.

P values	Day 4	Day 7	Day 14
No capsules	0.00001	0.03678	0.97188
Living non-IPN	0.01211	0.60405	0.91718
Pure Bla	0.00255	0.46766	0.99045

18. I don't understand the timescales in Figure 4. The authors say that the capsules were given before the ampicillin was given. The capsules only stayed in the gut for 4 to 5 hours, but the ampicillin was given for 3 days and the experiment run for 14 days. This doesn't make sense as antibiotic would be given after all the capsules had gone. Can this be clarified?

We thank the reviewer for the careful examination. On **every day** of days 1-3, capsules (25,000 capsules for each mouse) were fed (by oral gavage) to mice of groups 2-4 one hour before 5 mg of ampicillin was injected into each mouse through the tail vein (capsules were also fed for 3 days). We have further clarified this point in the revised text and methods in SI.

We thank the reviewer again for evaluating our manuscript and providing instructive comments.

Reference

- 1 Dragan, E. S. Design and applications of interpenetrating polymer network hydrogels. A review. *Chemical Engineering Journal* **243** 572–590 (2014).
- 2 Roland, C. M. in *Encyclopedia of Polymeric Nanomaterials* (ed Shiro Kobayashi Klaus Müllen) (Springer, Berlin, Heidelberg, 2015).
- 3 Lohani, A., Singh, G., Bhattacharya, S. S. & Verma, A. Interpenetrating polymer networks as innovative drug delivery systems. *J Drug Deliv* **2014**, 583612, doi:10.1155/2014/583612 (2014).
- 4 Schwartz, M. *Encyclopedia of Smart Materials*. (John Wiley & Sons, Inc. , 2002).
- 5 Sabu Thomas, D. G., Uros Cvelbar, K. V. S. N. Raju, Ramanuj Narayan, Selvin P. Thomas, Akhina H. *Micro- and Nano-Structured Interpenetrating Polymer Networks: From Design to Applications*. (John Wiley & Sons, Inc. , 2016).
- 6 Xin Chen, W. L., Wei Zhong, Yuhua Lu, Tongyin Yu. pH Sensitivity and Ion Sensitivity of Hydrogels Based on Complex-Forming Chitosan/Silk Fibroin Interpenetrating Polymer Network. *Journal of Applied Polymer Science* **65**, 2257–2262 (1997).
- 7 Pescosolido, L. *et al.* In situ forming IPN hydrogels of calcium alginate and dextran-HEMA for biomedical applications. *Acta Biomater* **7**, 1627-1633, doi:10.1016/j.actbio.2010.11.040 (2011).
- 8 Zhang, J. & Peppas, N. A. Synthesis and characterization of pH- and temperature-sensitive poly(methacrylic acid)/poly(N-isopropylacrylamide) interpenetrating polymeric networks. *Macromolecules* **33**, 102-107, doi:DOI 10.1021/ma991398q (2000).
- 9 Yuka Yamazawa, H. K., Tadashi Nakaji-Hirabayashi, Chiaki Yoshikawa, Hiromi Kitano, Kohji Ohno, Yoshiyuki Saruwatari and Kazuyoshi Matsuoka. Bioinactive semi-interpenetrating network gel layers: zwitterionic polymer chains incorporated in a cross-linked polymer brush. *Journal of Materials Chemistry B* **7**, 4280-4291 (2019).
- 10 Haque, M. A., Kurokawa, T. & Gong, J. P. Super tough double network hydrogels and their application as biomaterials. *Polymer* **53**, 1805-1822, doi:10.1016/j.polymer.2012.03.013 (2012).
- 11 Tomoki Ogoshi, H. I., Kyung-Min Kim, and Yoshiki Chujo. Synthesis of Organic–Inorganic Polymer Hybrids Having Interpenetrating Polymer Network Structure by Formation of Ruthenium–Bipyridyl Complex. *Macromolecules* **35**, 334–338 (2002).
- 12 Chen, A. *et al.* Synthesis and patterning of tunable multiscale materials with engineered cells. *Nature Materials* **13**, 515-523, doi:10.1038/NMAT3912 (2014).
- 13 Zhong, C. *et al.* Strong underwater adhesives made by self-assembling multi-protein nanofibres. *Nat Nanotechnol* **9**, 858-866, doi:10.1038/nnano.2014.199 (2014).
- 14 Chen, A. Y., Zhong, C. & Lu, T. K. Engineering Living Functional Materials. *Acs Synth Biol* **4**, 8-11, doi:10.1021/sb500113b (2015).
- 15 Le Feuvre, R. A. & Scrutton, N. S. A living foundry for Synthetic Biological Materials: A synthetic biology roadmap to new advanced materials. *Synth Syst Biotechnol* **3**, 105-112, doi:10.1016/j.synbio.2018.04.002 (2018).
- 16 Gilbert, C. & Ellis, T. Biological Engineered Living Materials: Growing Functional Materials with Genetically Programmable Properties. *ACS Synth Biol* **8**, 1-15, doi:10.1021/acssynbio.8b00423 (2019).
- 17 González, L. M., Mukhitov, N. & Voigt, C. A. Resilient living materials built by printing bacterial spores. *Nat Chem Biol* **16**, 126-133, doi:10.1038/s41589-019-0412-5 (2020).
- 18 Nguyen, P. Q., Courchesne, N. M. D., Duraj-Thatte, A., Praveschotinunt, P. & Joshi, N. S. Engineered Living Materials: Prospects and Challenges for Using Biological Systems to Direct the Assembly of Smart Materials. *Advanced Materials* **30**, doi:ARTN 170484710.1002/adma.201704847 (2018).
- 19 Praveschotinunt, P. *et al.* Engineered E. coli Nissle 1917 for the delivery of matrix-tethered therapeutic domains to the gut. *Nat Commun* **10**, 5580, doi:10.1038/s41467-019-13336-6 (2019).
- 20 Mohamad, N. R., Marzuki, N. H., Buang, N. A., Huyop, F. & Wahab, R. A. An overview of technologies for immobilization of enzymes and surface analysis techniques for immobilized enzymes. *Biotechnology, biotechnological equipment* **29**, 205-220, doi:10.1080/13102818.2015.1008192 (2015).
- 21 Homaei, A. A., Sariri, R., Vianello, F. & Stevanato, R. Enzyme immobilization: an update. *Journal of chemical biology* **6**, 185-205, doi:10.1007/s12154-013-0102-9 (2013).

- 22 Suo, H. *et al.* Interpenetrating polymer network hydrogels composed of chitosan and photocrosslinkable gelatin with enhanced mechanical properties for tissue engineering. *Mater Sci Eng C Mater Biol Appl* **92**, 612-620, doi:10.1016/j.msec.2018.07.016 (2018).
- 23 Jiandang Xue, L. L., Jiayou Liao, Yinghua Shen and Nanwen Li. Semi-interpenetrating polymer networks by azide-alkyne cycloaddition as novel anion exchange membranes. *Journal of Materials Chemistry A* **6**, 11317–11326 (2018).
- 24 Fernández-Gutiérrez, M. *et al.* Stimuli-responsive chitosan/poly (N-isopropylacrylamide) semi-interpenetrating polymer networks: effect of pH and temperature on their rheological and swelling properties. *J Mater Sci Mater Med* **27**, 109, doi:10.1007/s10856-016-5719-0 (2016).
- 25 Young A Han, E. M. L., and Byung Chul Ji. Mechanical Properties of Semi-interpenetrating Polymer Network Hydrogels Based on Poly(2-hydroxyethyl methacrylate) Copolymer and Chitosan. *Fibers and Polymers* **9**, 393-399 (2008).
- 26 Gross, L. A., Baird, G. S., Hoffman, R. C., Baldrige, K. K. & Tsien, R. Y. The structure of the chromophore within DsRed, a red fluorescent protein from coral. *Proc Natl Acad Sci U S A* **97**, 11990-11995, doi:10.1073/pnas.97.22.11990 (2000).
- 27 Jonassen, H., Kjoniksen, A. L. & Hiorth, M. Effects of ionic strength on the size and compactness of chitosan nanoparticles. *Colloid and Polymer Science* **290**, 919-929, doi:10.1007/s00396-012-2604-3 (2012).
- 28 Gu, Z. *et al.* Glucose-Responsive Microgels Integrated with Enzyme Nanocapsules for Closed-Loop Insulin Delivery. *ACS Nano* **7**, 6758-6766, doi:10.1021/nn401617u (2013).
- 29 Dai, Z. J. *et al.* Versatile biomanufacturing through stimulus-responsive cell-material feedback. *Nature Chemical Biology* **15**, 1017-+ (2019).
- 30 Marguet, P., Tanouchi, Y., Spitz, E., Smith, C. & You, L. Oscillations by Minimal Bacterial Suicide Circuits Reveal Hidden Facets of Host-Circuit Physiology. *Plos One* **5**, doi:10.1371/journal.pone.0011909 (2010).
- 31 Lee, K. Y. & Mooney, D. J. Alginate: Properties and biomedical applications. *Progress in Polymer Science* **37**, 106-126, doi:10.1016/j.progpolymsci.2011.06.003 (2012).
- 32 Mergulhão, F. J., Summers, D. K. & Monteiro, G. A. Recombinant protein secretion in Escherichia coli. *Biotechnol Adv* **23**, 177-202, doi:10.1016/j.biotechadv.2004.11.003 (2005).
- 33 Green, E. R. & Meccas, J. Bacterial Secretion Systems: An Overview. *Microbiol Spectr* **4**, doi:10.1128/microbiolspec.VMBF-0012-2015 (2016).
- 34 Lopez-Leon, T., Carvalho, E., Seijo, B., Ortega-Vinuesa, J. & Bastos-Gonzalez, D. Physicochemical characterization of chitosan nanoparticles: electrokinetic and stability behavior. *Journal of Colloid and Interface Science* **283**, 344-351, doi:10.1016/j.jcis.2004.08.186 (2005).
- 35 Enjalbert, B., Millard, P., Dinclaux, M., Portais, J. C. & Letisse, F. Acetate fluxes in Escherichia coli are determined by the thermodynamic control of the Pta-AckA pathway. *Sci Rep-Uk* **7**, doi:ARTN 4213510.1038/srep42135 (2017).
- 36 Kram, K. E. & Finkel, S. E. Rich Medium Composition Affects Escherichia coli Survival, Glycation, and Mutation Frequency during Long-Term Batch Culture. *Applied and Environmental Microbiology* **81**, 4442-4450, doi:10.1128/Aem.00722-15 (2015).
- 37 Karnaushenko, D. *et al.* Monitoring microbial metabolites using an inductively coupled resonance circuit. *Sci Rep-Uk* **5**, doi:ARTN 1287810.1038/srep12878 (2015).

REVIEWER COMMENTS

Reviewer #1 (Remarks to the Author):

The authors have adequately addressed my comments. I recommend its publication.

Reviewer #2 (Remarks to the Author):

The authors for the most part have addressed my concerns - the response regarding 'hard filler vs IPN' was lacking in my opinion. However, enough additional data was provided to make this a valid conclusion. I would ask the authors to do a rigorous check for grammar in their revised sections as I found numerous errors. All in all, interesting study and should be well-received by a broad community.

Reviewer #3 (Remarks to the Author):

Many aspects of the paper have been improved in the latest rounds of revisions. I applaud the authors for making the required changes and significantly improving the work and adding clarity to the many areas that were unclear originally. I did not notice that I wasn't alone among the reviewers in being unconvinced that the bacteria are producing a network of proteins upon lysing in the spheres. This has led to the authors now rebranding their materials as semi-IPNs for some sections of the paper (I note that IPN is still used in many figures). I don't personally know enough about materials science terminology to get into this argument about IPNs and sIPNs so I'll trust the authors on this. However, I still can't shake the hunch that what is actually happening here is just a lot of protein aggregates forming upon E.coli cell lysis. These aggregate clumps of protein are then just trapped in the network. I raised this possibility briefly in my first review but the reply from the authors seems to side-step it. I was hoping that TEM images with metal-stained proteins would be convincing enough to show the protein 'network' among the chitosan network but I have to say that I find Figure S7 very disappointing: it's very hard to see what is what (some labels on this figure would help) and it could easily just be showing the hallmarks of protein aggregates, rather than polymerised proteins forming a network.

I noticed in the rebuttal to one of the other reviewers that the proteins don't actually have to be a 'network' for it to be classed as an sIPN - they just need to polymerise, so I guess they could just do that in lump aggregates. If that's true then I think it needs to be made more clear up front in this paper. Already the majority of reviewers who have read this work have seen the title and expected to find proteins made from the E. coli forming a network. This will be the same with the readers, and will only disappoint a lot of readers if it turns out it is only a 'network' due to a technical definition, when it could easily just be polymerisation of protein into aggregate lumps - something that might not even be dependent on the SpyTag system.

With this in mind I think the paper still needs the following to be clarified and corrected.

1. Can the authors demonstrate that the protein has not just polymerised into aggregate lumps? I think this needs to be clearly established, and if it cannot then it should be discussed as a possibility in the paper.
2. The title should be altered to "Living fabrication of functional semi interpenetrating polymeric materials"

Also, is the labelling the wrong way around in Figure S6? The non-IPN images look more porous (they have larger pores) but this is not what is described

RESPONSE TO REVIEWERS

Below, we provide point-by-point responses to reviewers' comments (*italicized, blue*). The revision in the manuscript is highlighted in blue.

Reviewer #1 (*Remarks to the Author*):

The authors have adequately addressed my comments. I recommend its publication.

We thank the reviewer again for providing constructive comments.

Reviewer #2 (*Remarks to the Author*):

The authors for the most part have addressed my concerns - the response regarding 'bard filler vs IPN' was lacking in my opinion. However, enough additional data was provided to make this a valid conclusion. I would ask the authors to do a rigorous check for grammar in their revised sections as I found numerous errors. All in all, interesting study and should be well-received by a broad community.

We thank the reviewer again for providing the constructive comments. We have thoroughly checked our grammar in the manuscript to eliminate grammatical errors.

Reviewer #3: Reviewer #3 (*Remarks to the Author*):

Many aspects of the paper have been improved in the latest rounds of revisions. I applaud the authors for making the required changes and significantly improving the work and adding clarity to the many areas that were unclear originally. I did not notice that I wasn't alone among the reviewers in being unconvinced that the bacteria are producing a network of proteins upon lysing in the spheres. This has led to the authors now rebranding their materials as semi-IPNs for some sections of the paper (I note that IPN is still used in many figures). I don't personally know enough about materials science terminology to get into this argument about IPNs and sIPNs so I'll trust the authors on this.

We thank the reviewer for recognizing that the revised manuscript has been improved by incorporating new results and clarifications.

However, I still can't shake the hunch that what is actually happening here is just a lot of protein aggregates forming upon E.coli cell lysis. These aggregate clumps of protein are then just trapped in the network. I raised this possibility briefly in my first review but the reply from the authors seems to side-step it. I was hoping that TEM images with metal-stained proteins would be convincing enough to show the protein 'network' among the chitosan network but I have to say that I find Figure S7 very disappointing: it's very hard to see what is what (some labels on this figure would help) and it could easily just be showing the hallmarks of protein aggregates, rather than polymerised proteins forming a network.

We thank the reviewer for the constructive comment. We have added the labels on Figure S7 to better identify the location and arrangement of gold-nanoparticles on sections of sIPN and non-IPN capsules. Our results show the polymerized protein structure in living sIPN capsules as indicated by the densely alignment of the gold nanoparticles. In comparison, there is only sparse and scattered distribution of gold nanoparticles in the non-IPN capsules.

I noticed in the rebuttal to one of the other reviewers that the proteins don't actually have to be a 'network' for it to be classed as an sIPN - they just need to polymerise, so I guess they could just do that in lump aggregates. If that's true then I think it needs to be made more clear up front in this paper. Already the majority of reviewers who have read this work have seen the title and expected to find proteins made from the E. coli forming a network. This will be the same with the readers, and will only disappoint a lot of readers if it turns out it is only a 'network' due to a technical definition, when it could easily just be polymerisation of protein into aggregate lumps - something that might not even be dependent on the SpyTag system.

We thank the reviewer for this constructive comment.

Indeed, it is important for us to better clarify the concept of the interpenetrating polymer network (IPN) to avoid any potential confusion. An IPN is defined as a blend of two or more polymers in a network with at least one of the systems synthesized in the presence of another. Two categorizing schemes are used to describe IPNs¹⁻⁵. Based on the structure, IPNs are classified into semi- and full IPNs, depending on whether one or both of the respective components are fully crosslinked⁶⁻⁸. Based on method of synthesis, IPNs are classified into sequential or simultaneous IPNs. In a sequential IPN, the first crosslinked polymer network is swollen by the monomer of the second polymer that is polymerized and/or cross-linked afterwards^{9,10}. While in a simultaneous IPN, both components are polymerized concurrently¹¹.

In our work, we demonstrate the engineering of living materials consisting of programmed bacteria and non-living components. The fabrication process is driven by the engineered bacteria encapsulated in a polymeric microcapsule (*crosslinked network*), which serves as the initial scaffold (Figure S2). The bacteria grow and undergo programmed lysis in a density-dependent manner, releasing protein monomers decorated with reactive tags (Figure 2A). Those protein monomers polymerize with each other to form the second polymeric component that is interlaced with the initial crosslinked scaffold (Figure 2, S4 and S7). Our results support these characteristics described above and they are fully consistent with the definition of semi-IPN. **In light of reviewers' comments and to avoid potential confusion, we have further clarified that the protein is polymerized but not necessarily fully crosslinked (in the Discussion section).**

In the current work, the protein monomers functionalized with reactive tags (SpyTag-SpyCatcher) can polymerize into high molecular weight product (>460kDa) at a monomer concentration mimicking the capsules interior (Figure S4). We used Spy-chemistry since SpyTag-SpyCatcher proceeds efficiently at temperatures ranging from 4 to 37 °C and does not require additional chemical reagents or catalysts. The reviewer is correct that the whole engineering strategy is not limited to the use of SpyTag/SpyCatcher. We have further clarified this part in the discussion, as highlighted by the blue color.

With this in mind I think the paper still needs the following to be clarified and corrected.

1. Can the authors demonstrate that the protein has not just polymerised into aggregate lumps? I think this needs to be clearly established, and if it cannot then it should be discussed as a possibility in the paper.

We thank the reviewer for this suggestion. In our previous revision, we have made it clear that the protein monomers polymerize with each other to form the second polymeric component that is interlaced with the initial crosslinked scaffold (Figure 1, abstract, introduction and results). As the reviewer suggested, we have further clarified that the protein monomers are polymerized but not necessarily fully crosslinked into network in the Figure 1 and discussion part, as highlighted by the blue color. As noted above, our use of the term semi-IPN is fully consistent with what is used in the IPN literature.

2. The title should be altered to "Living fabrication of functional semi interpenetrating polymeric materials"

We thank the reviewer for this suggestion. We have altered the title to "Living fabrication of functional semi-interpenetrating polymeric materials".

Also, is the labelling the wrong way around in Figure S6? The non-IPN images look more porous (they have larger pores) but this is not what is described.

We thank the reviewer for the very careful examinations. The labelling of the Figure S6 is correct. The reviewer is correct that the non-IPN has large pores. Porosity is defined as the fraction of the volume of voids over the total volume. We estimated that the sIPN has more volume of voids since the total volume of packed smaller pores is bigger than that of the packed large pores (assuming the total volume is equivalent).

To avoid potential confusion, we have revised our texts and compared them in terms of pore size, as highlighted by the blue color.

References

- 1 Dragan, E. S. Design and applications of interpenetrating polymer network hydrogels. A review. *Chemical Engineering Journal* **243** 572–590 (2014).
- 2 Roland, C. M. in *Encyclopedia of Polymeric Nanomaterials* (ed Shiro Kobayashi Klaus Müllen) (Springer, Berlin, Heidelberg, 2015).
- 3 Lohani, A., Singh, G., Bhattacharya, S. S. & Verma, A. Interpenetrating polymer networks as innovative drug delivery systems. *J Drug Deliv* **2014**, 583612, doi:10.1155/2014/583612 (2014).
- 4 Schwartz, M. *Encyclopedia of Smart Materials*. (John Wiley & Sons, Inc. , 2002).
- 5 Sabu Thomas, D. G., Uros Cvelbar, K. V. S. N. Raju, Ramanuj Narayan, Selvin P. Thomas, Akhina H. *Micro- and Nano-Structured Interpenetrating Polymer Networks: From Design to Applications*. (John Wiley & Sons, Inc. , 2016).
- 6 Xin Chen, W. L., Wei Zhong, Yuhua Lu, Tongyin Yu. pH Sensitivity and Ion Sensitivity of Hydrogels Based on Complex-Forming Chitosan/Silk Fibroin Interpenetrating Polymer Network. *Journal of Applied Polymer Science* **65**, 2257–2262 (1997).
- 7 Pescosolido, L. *et al.* In situ forming IPN hydrogels of calcium alginate and dextran-HEMA for biomedical applications. *Acta Biomater* **7**, 1627-1633, doi:10.1016/j.actbio.2010.11.040 (2011).
- 8 Zhang, J. & Peppas, N. A. Synthesis and characterization of pH- and temperature-sensitive poly(methacrylic acid)/poly(N-isopropylacrylamide) interpenetrating polymeric networks. *Macromolecules* **33**, 102-107, doi:DOI 10.1021/ma991398q (2000).
- 9 Yuka Yamazawa, H. K., Tadashi Nakaji-Hirabayashi, Chiaki Yoshikawa, Hiromi Kitano, Kohji Ohno, Yoshiyuki Saruwatari and Kazuyoshi Matsuoka. Bioinactive semi-interpenetrating network gel layers: zwitterionic polymer chains incorporated in a cross-linked polymer brush. *Journal of Materials Chemistry B* **7**, 4280-4291 (2019).
- 10 Haque, M. A., Kurokawa, T. & Gong, J. P. Super tough double network hydrogels and their application as biomaterials. *Polymer* **53**, 1805-1822, doi:10.1016/j.polymer.2012.03.013 (2012).
- 11 Tomoki Ogoshi, H. I., Kyung-Min Kim, and Yoshiki Chujo. Synthesis of Organic–Inorganic Polymer Hybrids Having Interpenetrating Polymer Network Structure by Formation of Ruthenium–Bipyridyl Complex. *Macromolecules* **35**, 334–338 (2002).

REVIEWER COMMENTS

Reviewer #3 (Remarks to the Author):

I am satisfied with the revisions and think this paper is suitable for publication now.

Below, we provide point-by-point responses to reviewers' comments (*italicized, blue*). The revision in the manuscript is highlighted in blue.

Reviewer #3 (*Remarks to the Author*):

I am satisfied with the revisions and think this paper is suitable for publication now.

We thank the reviewer again for providing constructive comments.